# A spatially localized DNA linear classifier for cancer diagnosis

Linlin Yang[1,2,3,5], Qian Tang[1,5], Mingzhi Zhang [2], Yuan Tian[2], Xiaoxing Chen[2], Rui Xu[4], Qian Ma[4], Pei Guo [1] ✉, Chao Zhang [2,4] ✉ & Da Han [1,2] ✉

Molecular computing is an emerging paradigm that plays an essential role in data storage, bio-computation, and clinical diagnosis with the future trends of more efficient computing scheme, higher modularity with scaled-up circuity and stronger tolerance of corrupted inputs in a complex environment. Towards these goals, we construct a spatially localized, DNA integrated circuits-based classifier (DNA IC-CLA) that can perform neuromorphic architecture-based computation at a molecular level for medical diagnosis. The DNA-based classifier employs a two-dimensional DNA origami as the framework and localized processing modules as the in-frame computing core to execute arithmetic operations (e.g. multiplication, addition, subtraction) for efficient linear classification of complex patterns of miRNA inputs. We demonstrate that the DNA IC-CLA enables accurate cancer diagnosis in a faster (about 3 h) and more effective manner in synthetic and clinical samples compared to those of the traditional freely diffusible DNA circuits. We believe that this all-in-one DNA-based classifier can exhibit more applications in bio-computing in cells and medical diagnostics.

As an important algorithm in machine learning, classification implemented by classifiers can be trained to sort complex data patterns with accuracy and predictivity[1,2]. Given the massive amounts of data from multi-omics, such as genomics, transcriptomics, proteomics, and metabolomics, diagnostic classifiers are developing quickly and have greatly improved the accuracy, efficiency, and effectiveness of medical diagnosis, ultimately leading to better health outcomes for patients[3–13]. For instance, plasma cell-free RNA-based classifiers have been used to determine the risk of developing pre-eclampsia months before clinical presentation[9,10]. Cell-free DNA and DNA methylation maps have also been involved to build classifiers for early diagnosis and screening of various cancer subtypes and stages[11].

Synthetic DNA is a programmable polymer that can store and process information; therefore, it has been engineered for versatile computational devices ranging from logic circuits to neural networks[14–17]. Plus, DNA also provides a natural interface for molecular recognition (e.g. DNA, RNA, proteins, and metabolic molecules) that is the key to creating complex biological input patterns[18–20]. Therefore, DNA-based molecular classifiers have found their niche in implementing the in silico trained classifiers for powerful pattern recognition and accurate disease diagnosis[21–29]. For instance, simple Boolean logic gates have been designed on cell membranes for accurate cancer cell classification and intelligently controlled drug delivery[21–25,30]. DNA-based neural networks relying on linear classification mechanisms have also been developed for the classification of letters in the MNIST database and special languages such as Chinese oracles[26,31]. Similar DNA-based linear classifiers have been further expanded for disease identification, such as etiological and cancer diagnosis[27–29]. Recently, a DNA-enzyme hybrid-based classifier with tunable weights and biases has been designed to perform non-linear decision-making in response

[1]Zhejiang Cancer Hospital, Hangzhou Institute of Medicine (HIM), Chinese Academy of Sciences, 310022 Hangzhou, Zhejiang, China. [2]Institute of Molecular Medicine, Shanghai Key Laboratory for Nucleic Acid Chemistry and Nanomedicine, Renji Hospital, School of Medicine, Shanghai Jiao Tong University, 200127 Shanghai, China. [3]School of Pharmacy, Shandong Technology Innovation Center of Molecular Targeting and Intelligent Diagnosis and Treatment, Binzhou Medical University, 264003 Yantai, China. [4]Intellinosis Biotech Co.Ltd., 201112 Shanghai, China. [5]These authors contributed equally: Linlin Yang, Qian Tang. ✉e-mail: guopei@ibmc.ac.cn; chaozhang@sjtu.edu.cn; dahan@sjtu.edu.cn

to microRNA (miRNA) inputs with a higher classification sensitivity and speed[32]. Overall, these showcased the trends of DNA-based classifiers: faster response with neuromorphic architecture over Boolean types, higher parallelism with scaled-up circuitry, as well as stronger tolerance of corrupted inputs in a complex environment.

However, current DNA-based classifiers rely exclusively on interactions between diffusible molecular components, which are limited by relatively slow kinetics and complex design of probes with high sequence specificity for orthogonal reactions[15]. These issues greatly increase the difficulty in designing larger-sized circuits and impact the error-correcting capability of the classifiers. Embedding information-

processing circuitry in a spatially localized architecture can be a superior route to improve computation efficiency with reduced design complexity. The adaptable design of DNA structures, especially DNA origami, harnesses their addressable and programmable characteristics to enable the fine-tuning of stoichiometry, the segregation of distinct functional modules, and consequently, serve as ideal scaffolds for a diverse array of DNA reaction networks[33–37]. So far, the intelligent levels of reported localized DNA devices are mainly Boolean-based structures that can only respond to the presence or absence of targets. There is still a lack of neuromorphic architecture-based localized molecular classifiers that can process complex input patterns and

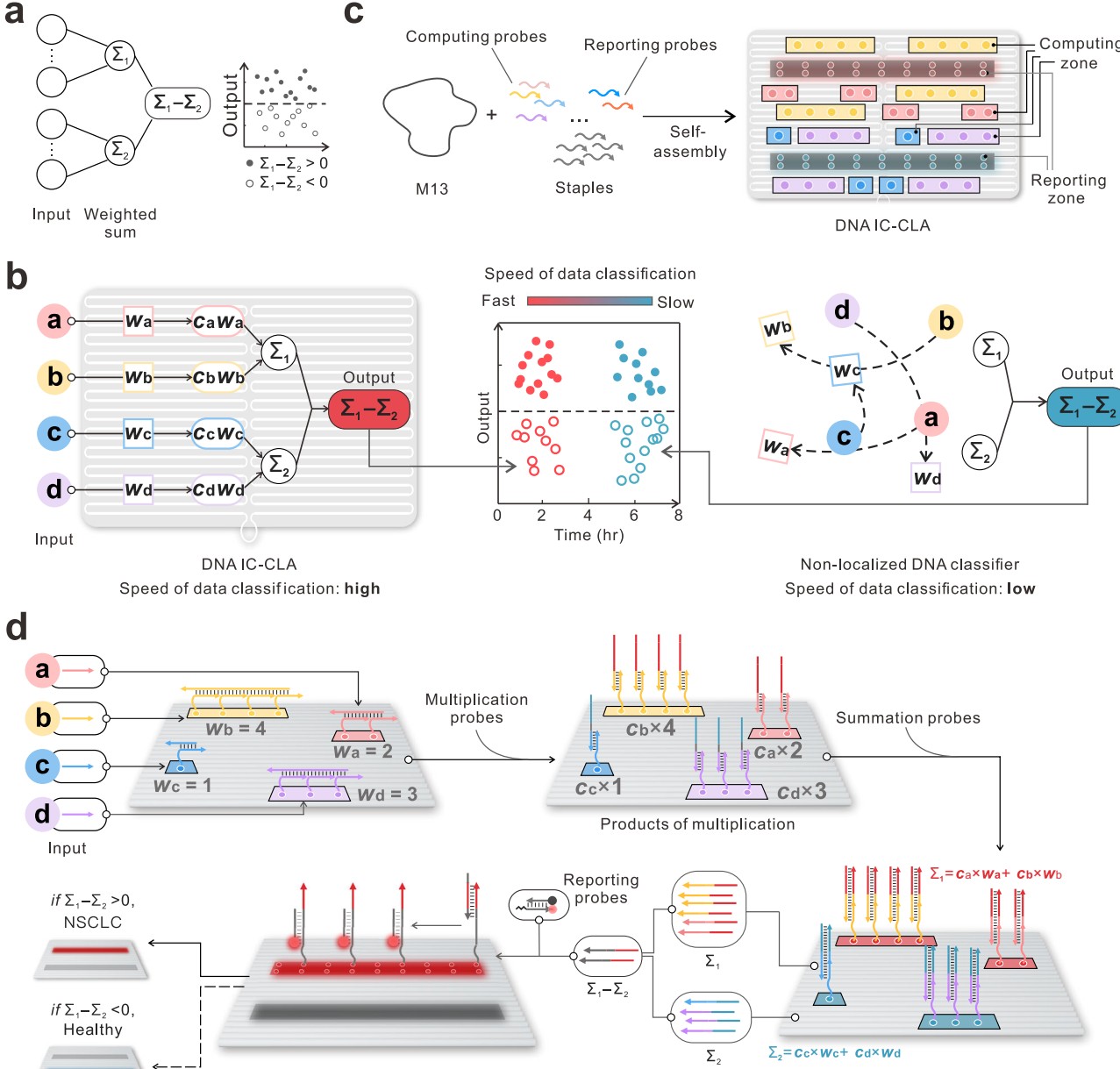

**Fig. 1 | DNA IC-CLA for cancer diagnosis. a** Neuromorphic architecture-based scheme for linear classification. **b** Scheme illustrating DNA IC-CLA and non-localized DNA classifier with different speeds of classification. **c** Design and assembly scheme for DNA IC-CLA that consists of a rectangular-shaped breadboard, multiple computing cores for arithmetic operations, and reporting zones for classification of diagnostic results. The computing zone contains four different spatially separated zones (four identical replications) labeled with different colors for analyzing different inputs with programmable arithmetic operations. The reporting zone contains two different areas with 18 docking sites modified with fluorophores (FAM and ROX) or biotin-labeled probes in each line for reporting different classification results. **d** Detailed scheme for the entire classification process of cancer diagnosis with miRNAs as inputs. NSCLC: non-small cell lung cancer. For a classic linear classification equation with arithmetic operations of multiplication, summation, and subtraction, the computing cores on DNA IC-CLA can be programed and cascaded by DNA strand displacement reactions.

respond to different concentrations of targets in biological samples with accurate classification results, which further limits the localized molecular classifiers for applications in biocomputing and medical diagnostics.

In this work, we decoded the in silico-trained classifier to a spatially localized, DNA *i*ntegrated *c*ircuits-based *cla*ssifier (DNA IC-CLA) at a molecular level. The computing core of DNA IC-CLA is a neuromorphic architecture that can execute arithmetic operations of the in silico-trained support vector machine (SVM)-based linear classifier, containing multiplication, summation, and subtraction, in response to multiple inputs (e.g., miRNAs) and execute classification in a programmable manner (Fig. 1a, b). We demonstrate that the DNA IC-CLA can perform cancer diagnosis in a much faster and more effective manner in synthetic and clinical samples without manual intervention. This system provides guidance for engineering localized molecular computing devices and further expands the power of DNA-based computing for complex contents, such as computation inside cells, liquid biopsies, and DNA storage.

## Results

### Design of DNA IC-CLA

Our DNA IC-CLA employs a two-dimensional DNA origami as the breadboard and localized cascaded DNA strand displacement reactions as the in-frame computing driving force in the computing and reporting modules (Fig. 1b, c). Herein, the DNA origami framework provides a programmable scaffold for the precise organization of DNA-based computing probes and a spatial constraint for accelerating proper interactions as well as reducing interference between different probes by separating them. On the other hand, linear classifiers typically need arithmetical operations such as multiplication, summation, and subtraction for classification. Therefore, the key becomes to design and immobilize computing elements for carrying out arithmetical operations on the origami framework. Here, we adopt a modular design with multiple localized cores on the DNA breadboard for carrying out different activation functions such as weighting, addition, and subtraction to the input concentrations. The nature of these arithmetical operations is to perform precise concentration-based transformation of molecular inputs with localized DNA-based displacement reactions for classification. Fig. 1c and Supplementary Fig. 1 show the layout of DNA IC-CLA, consisting of different probes for arithmetical operations and output reporting. We designed four different zones in total for specifically recognizing four different inputs, and each zone was replicated with four identical copies on the origami (90 × 60 nm). The multiplication, addition, and subtraction operations can be programed in each zone and cascaded by localized DNA strand displacement reactions (Fig. 1d). We first used gel electrophoresis and atomic force microscopy (AFM) imaging to characterize the successful assembly and modification of DNA IC-CLA. Agarose gel electrophoresis result demonstrates the formation of both DNA origami breadboard and DNA IC-CLA, as corroborated by the higher molecular weight of DNA IC-CLA upon successful immobilization of DNA computing and reporting probes (Supplementary Fig. 2a). AFM images also show the uniformly formed DNA IC-CLA with the desired size of 90 × 60 nm and the modified zones with higher heights (Supplementary Fig. 2b-d).

We then opted to validate the precise modifications of probes and the effective arithmetical operations in response to different concentrations of inputs that occurred on the localized origami surface. For the weighting operation with the formula of $c(\text{output})_n = w_n \times c(\text{input})_n$, where $w_n$ is the predefined weight, $c(\text{input})_n$ is the initial concentration of input, and $c(\text{output})_n$ is the transformed concentration of output, respectively, we designed computing probes ($L_n$–$N$) that pre-locked the corresponding output $c(\text{output})_n$ in the sequences of ssDNA (i.e., $A_1$, $A_2$...$A_n$ for $L_1$–$A$) on the surface of DNA origami. Upon the activation of input (a), $L_1$–$A$ (excess

to the input concentration) can be freed up to expose a precisely weighted amount of output (also proportional to the concentration of input) with the help of freely diffusible high concentration of helper strands ($H$ probes) through a concentration disequilibria-driven strand displacement reaction. In this way, different weighted results can be precisely encoded in $N$ probes (e.g. $A_1$, $A_2$...$A_n$ for Input a) and localized on the surface of DNA origami without diffusing away (Fig. 2a, Supplementary Figs. 3, 4). As shown in Fig. 2b, we used $M$ probes and a FAM-labeled $M$ reporter to exhibit the results of the weighting operations and found the conformity with the pre-defined weights of 1–4, respectively. We have conducted a titration of input concentrations ranging from 0 to 20 nM and observed a robust linear correlation between the fluorescence response and the concentration of Input b, particularly in the range of 5 nM to 20 nM. This observation underscores the efficacy of the molecular implementation of the weighting operations as described by the formula: $c(\text{output})_n = w_n \times c(\text{input})_n$ (as illustrated in Supplementary Fig. 5). However, in the lower concentration range of 1–5 nM, the linearity is less pronounced. This discrepancy may be attributed to factors such as instrumental inaccuracies or limitations in the sensitivity to low input concentrations. Besides, we investigated the background signal in multiplication operation and identified the source of leakage as primarily stemming from interactions between $M$ probes and $M$ reporters, which are freely diffusible at high concentrations and drive the cascaded strand displacement reaction (Supplementary Fig. 6). This issue could potentially be mitigated by refining the sequence design of the probes, such as by employing the 1 nt-gap design[38].

To experimentally implement the summation calculation on the surface of origami, we utilized the output ($w_n \times c(\text{input})_n$) obtained from multiplications as the input for this step. In specific, the weighted outputs ($a \times w_a$ and $b \times w_b$) on the origami from the last step are recognized by excess freely diffusible $S$ probes and summed up by forming the complex on the surface of DNA IC-CLA for the implementation of $a \times w_a + b \times w_b$ (Fig. 2c). We also used a fluorescent reporter ($S$ reporter) to report the summed fluorescence signals and found good agreement with the pre-designed scheme (Fig. 2d, Supplementary Figs. 3 and 7). Thus, we successfully established molecular implementation of the summation calculation for the same type of inputs on DNA IC-CLA.

We next designed and proved a spatially localized subtraction operation on this DNA IC-CLA. It is typically difficult to implement negative weights or weighted sums for DNA-based computation[15,39]. Here, we artificially assigned one type of weighted sum encoded in a specific DNA sequence as negative and adopted a winner-takes-all scheme to execute the subtraction calculation[40]. Specifically, a freely diffusible annihilator probe ($N$ probe) can cancel out identical amounts of positive and negative DNA weighted sums ($E$ and $F$ probes) and only leave the excess one with an active reporting sequence on the DNA origami surface. We immobilized two reporting probes ($RE$ and $RF$ probes) on the surface of DNA IC-CLA for more rapid output reporting, in which an entropy-driven catalytic amplification for the remaining probe was used for reporting and improving the sensitivity (Supplementary Fig. 8). The entropy-driven catalytic reactions involving $E$ and $RE$ probes, as well as $F$ and $RF$ probes, are initiated by Fuel $E$ and Fuel $F$, respectively. These fuels facilitate the release of $E$ and $F$, enabling them to participate in additional catalytic cycles. As shown in Fig. 2e, f, we demonstrated this process using different concentrations of weighted sum ($E$ and $F$ probes) combinations ranging from 0 to 50 nM (Fig. 2g and Supplementary Fig. 9). Overall, these results showed that only the weighted sums with higher concentrations resulted in higher signals, which is in good agreement with the simulation of subtraction operation. However, it should be noted that the signal differences between two weighted sums become noisy when the two concentrations of two inputs are similar. This observation has also been

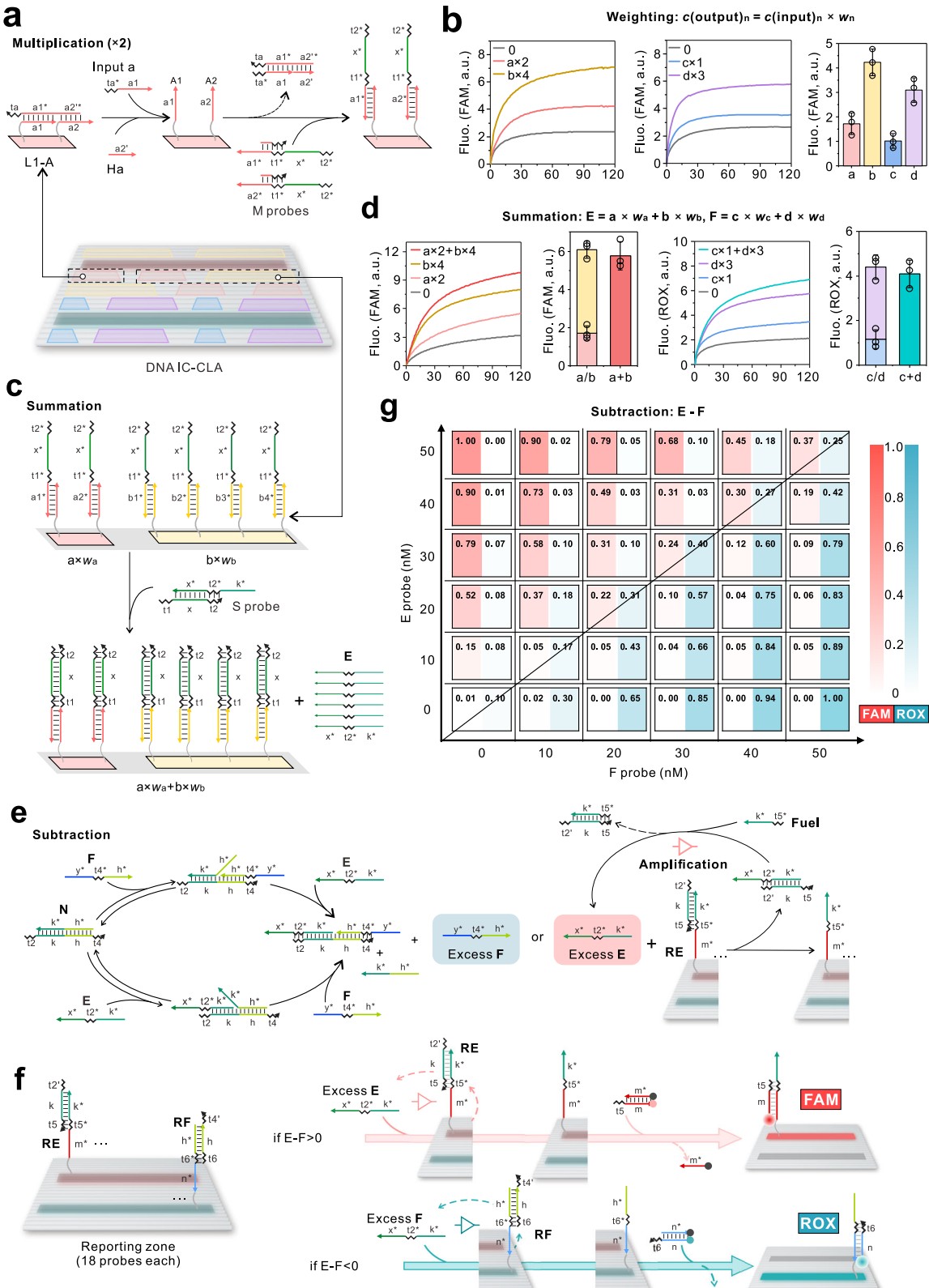

found in the previous report and indicates that DNA IC-CLA can perform accurate subtraction when two inputs have distinguishable concentrations[27].

We next experimentally compared the computing speed of DNA IC-CLA and the free diffusion system in response to the same concentrations of inputs. For three-step DNA strand displacement reactions (SDRs) of weighting and four-step SDRs of summation, DNA IC-

CLA showed slightly faster kinetics with 1–1.5 times higher reaction rate ($v_{max}$) (Fig. 3a). Taking into account that the reaction network yields products at an initial rate that is approximately linear in the first 10 min, which is closely mirroring the actual reaction rate of DNA computing cascades, the rate progressively decreases as the reaction continues and the concentrations of the probes fluctuate. The reaction rate ($v_{max}$) is defined as the slope of the fluorescence kinetics in the

**Fig. 2 | Design and validation of DNA IC-CLA. a** Scheme for multiplication operation $c(\text{output})_n = w_n \times c(\text{input})_n$ on DNA IC-CLA. Each $L_n$–$N$ can be recognized by corresponding input and release fixed numbers of pre-locked output $c(\text{output})_n$ equivalent to the pre-defined weight. As exemplified by $w_n = 2$, $M$ probe and $M$ reporter are used to report the multiplication results. Other situations with different weights are shown in Supplementary Figs. 3, 4. **b** Multiplication fluorescence kinetics of DNA IC-CLA with DNA inputs of different weights ($w_n = 1$–4). Fluorescence difference ($F_t$–$F_0$) at 60 min was used to compare individual signals. $F_t$: steady-state fluorescence. $F_0$: background fluorescence. **c** Scheme for summation ($a \times w_a + b \times w_b = E$; $c \times w_c + d \times w_d = F$) of DNA IC-CLA. Exemplified by $a \times w_a + b \times w_b = E$, the summation operation is executed by converting the same category of Input $a$ and $b$ into the complex on the origami and signal probe of $E$ by interacting with the $S$ probe. $S$ reporter verifies the summation results. Other summations ($c \times w_c + d \times w_d = F$) are listed in Supplementary Fig. 7. **d** Summation fluorescence kinetics of DNA IC-CLA with Input $a$ and $b$ alone and their sums in the presence of an equal concentration of $S$ probes and $S$ reporter (FAM), and with Input c and d alone and their sums in the presence of an equal concentration of $S$ probes and $S$ reporter (ROX). $F_t$–$F_0$ at 60 min was used to compare individual and summation signals. Data in **b** and **d** are presented as mean values ± SD, $n = 3$ biological replicates. **e, f** Scheme for subtraction ($E$–$F$ = diagnostic result) and amplification step for sensitivity improvement. The subtraction of remaining $E$ and $F$ probes are reported by FAM and ROX, respectively. The catalytic amplification process is listed in Supplementary Fig. 8. **g** Subtraction results for different combinations of $E$ and $F$. Fluorescence signal acquisition from the subtraction fluorescence kinetics (at 120 min) of DNA IC-CLA with different combinations of $E$ and $F$, normalized using a common minimum and maximum fluorescence level. The diagonal line indicates equal concentrations of both strands. Samples resulting in a signal of [FAM]−[ROX] > 0 or [FAM]−[ROX] < 0 are assigned to different classification groups. Source data are provided as a Source Data file.

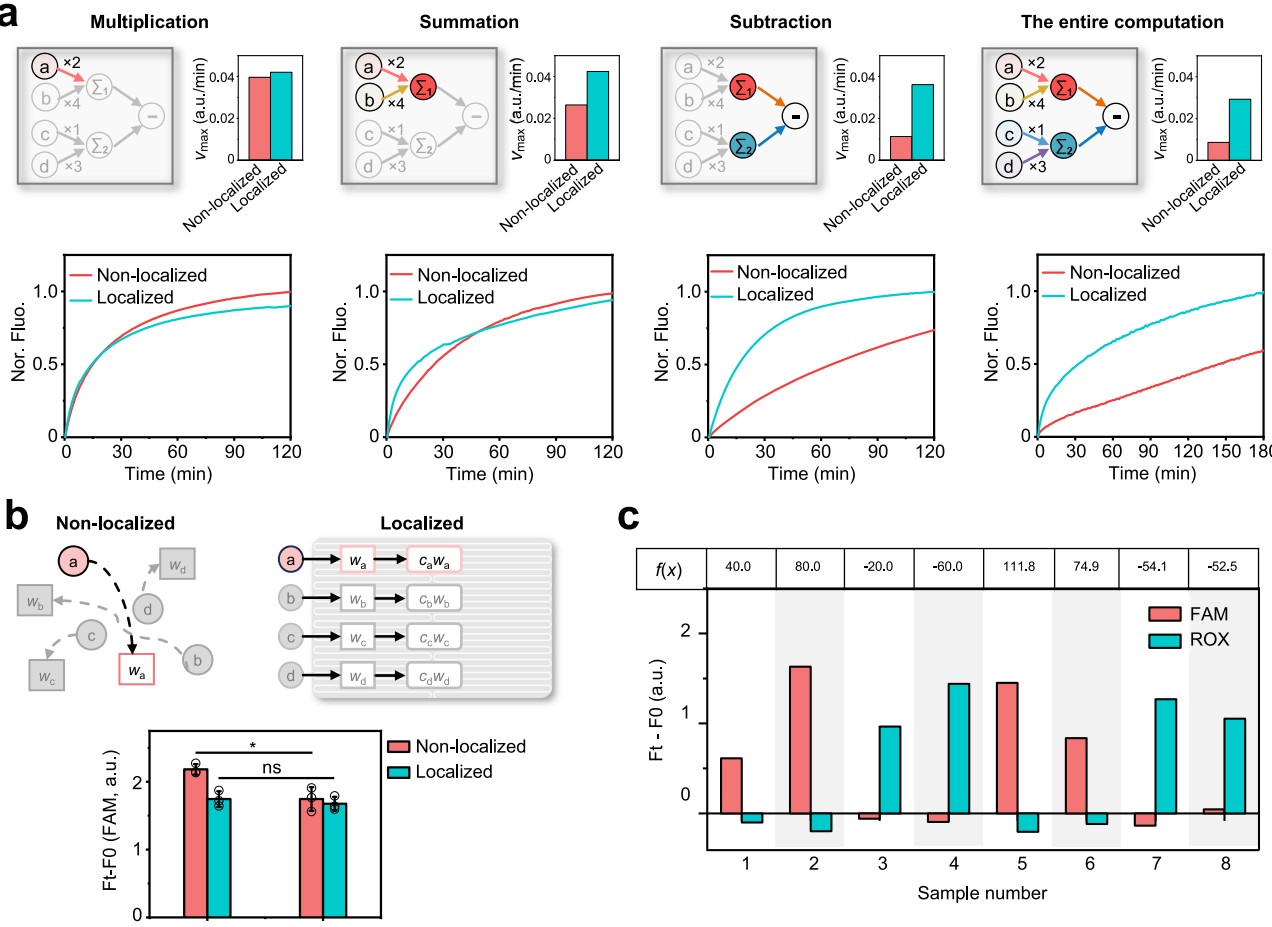

**Fig. 3 | Performance comparison between DNA IC-CLA and non-localized system. a** Scheme and fluorescence kinetics of multiplication, summation, subtraction, and entire computation of non-localized system and DNA IC-CLA. $v_{max}$ is the slope of fluorescence kinetics in the first 10 min. Multiplication reactions were carried out with 5 nM of DNA IC-CLA or 20 nM of each $L_n$–$N$, 25 nM of each $H$ probe, 25 nM of each $M$ probe, 200 nM of each $M$ reporter, and 20 nM of Input $a$. Summation reactions were carried out with 5 nM of DNA IC-CLA or 20 nM of each $L_n$–$N$, 25 nM of each $H$ probe, 25 nM of each $M$ probe, 150 nM of each $S$ probe, 200 nM of each $S$ reporter, and 20 nM of Input $a$ and Input $b$. Subtraction reactions were carried out with 5 nM of DNA IC-CLA or 20 nM of each $L_n$–$N$, 200 nM of $N$ probe, 200 nM of each Fuel probe (Fuel $E$ and Fuel $F$), 200 nM of each reporter, 20 nM of $E$ probe and 50 nM $F$ probe. The entire computations were carried out with 5 nM of DNA IC-CLA or 20 nM of each $L_n$–$N$, 25 nM of each $H$ probe, 25 nM of each $M$ probe, 150 nM of each $S$ probe, 200 nM of $N$ probe, 200 nM of each Fuel probe (Fuel $E$ and Fuel $F$), 200 nM of each reporter, and 20 nM of each Input. **b** Scheme and fluorescence difference ($F_t$–$F_0$) of multiplication of non-localized system and DNA IC-CLA under different preparation times. Data are presented as mean values ± SD, $n = 3$ biological replicates. Statistical analysis by two-tailed unequal variance $t$-test, *$P < 0.05$. Exact $P$-values are provided in the Source Data file. **c** Classification results of DNA IC-CLA for varying combinations of DNA inputs. The activation of linear classifier was set to be $f(x) = 2 \times c(a) + 4 \times c(b) - 1 \times c(c) - 3 \times c(d)$. Fluorescence difference ($F_t$–$F_0$) of eight samples with different combinations of DNA input concentrations was plotted against sample numbers (Nos. 1–8). The table shows the theoretical values of $f(x)$. Source data are provided as a Source Data file.

first 10 min. When more steps of computation, such as subtraction (over 5 steps of SDRs), were cascaded, the localized DNA IC-CLA showed a much higher $v_{max}$ (3 times higher). To demonstrate the computational speedup achieved by localizing computing probes on our DNA-IC CLA, we have conducted simulations comparing non-localized and localized scenarios for multiplication, summation, and subtraction operations individually (as detailed in Supplementary Note 1 and Supplementary Fig. 10), with the simulation results consistent with the experimental observation. Next, we tested the completion time of an intact activation function of $f(x) = 2 \times c(a) + 4 \times c(b) -1 \times c(c) -3 \times c(d)$ with these two systems that contained six steps of cascading DNA-based strand displacement reactions and four different inputs. The DNA IC-CLA exhibited a completion computing time of 1.3 h, while the freely diffusible system needed more than 3 h to show the result. The $v_{max}$ between the two systems also exhibited a more than 3 times difference with the higher speed represented by DNA IC-CLA. It should be noticed that this DNA IC-CLA can perform the linear classification even faster than the enzyme-based computation scheme for a similar type of activation function[32] which greatly highlights the superiority of localizing DNA-based elements for computing speed improvement. Other than benefiting from the higher computing speed resulting from higher local concentrations of reactants, our modular designed system also exhibits higher robustness as the effective arithmetic operations (e.g. multiplication) could still be accurately performed after 1 week post the assembly of DNA IC-CLA, while the free diffusion system lost computation accuracy after 1 week post the mixing of all components (Fig. 3b and Supplementary Fig. 11). One reason is that the localization of DNA strands on surface places a spatial constraint for preventing unwanted collision between freely diffusible strands. The other reason is believed to be the higher biostability originated from the entire assembled nanostructures over small ssDNA and dsDNA[41]. Overall, we found that localized DNA IC-CLA is faster and more robust than the free diffusion-based system in performing classifications.

A successful linear classifier requires cascaded and accurate mathematical operations for effective data classification[15]. Supplementary Fig. 12 indicates that the DNA IC-CLA's adherence to the pre-established weights of 1–4 deviated slightly from the calibration system, which stipulated that the fluorescence signal of the sample with the addition of Input $c$ should be set to 1 to calibrate the fluorescence signals of the remaining three samples. One possible source of calculation errors may originate from an incomplete modification of computing probes (e.g. $L_n$–$N$) on the origami. Therefore, we performed multiple rounds of pre-separation of DNA IC-CLA with a purification column to minimize the modification error. We next tested the classification performance of DNA IC-CLA with different combinations of inputs. As shown in Fig. 3c and Supplementary Fig. 13, our DNA IC-CLA is sensitive to nanomolar inputs and able to classify input profiles accurately by correctly reporting positive (FAM) and negative (ROX) signals. Furthermore, we tested 40 samples to determine the classification margin of DNA IC-CLA (Supplementary Note 2, Supplementary Fig. 14) and found samples can be correctly classified when the values of $f(x) = E - F$ are in the range of $f(x) > 10$ nM or $f(x) < -10$ nM. In addition, we also used different reporting methods to consolidate the correct computation results of DNA IC-CLA. We put 18 docking sites for two different biotin-labeled reporters at the defined positions on the origami. Only the completion of correct sequential computation can result in the labeling of DNA reporters with biotins in the right locations, followed by visualization with streptavidin markers by AFM. Supplementary Fig. 15 demonstrates that the DNA IC-CLA is capable of executing accurate computations with streptavidin markers positioned as anticipated when either Input a or Input c is present. This evidence affirms our DNA IC-CLA as a neuromorphic architecture integrated into a physical chip, where multiple artificial neurons are defined by their locations and pre-determined weights and threshold

values. This design enables the potential to construct more sophisticated, high-speed nano-scale neuromorphic architectures with enhanced functionalities.

### In silico-trained classifier model and implementation

After verification of the effective classification with different combinations of inputs by DNA IC-CLA, we attempted to apply it for disease diagnostics. Here, we performed SVM-model training using miRNA-seq expression profiles of healthy individuals and non-small cell lung cancer (NSCLC) patients publicly available in The Cancer Genome Atlas (TCGA)[42]. Specifically, we first conducted differential expression gene analysis to obtain 33 up-regulated and 22 down-regulated miRNAs as potential biomarkers to train the classification model (Supplementary Fig. 16a). Then, we set constraints on the input miRNAs such as the number of biomarkers and weight scopes, and performed SVM training to obtain the optimal linear classification model as: diagnostic result = $[c(\text{miR-148a-3p}) \times 2 + c(\text{miR-182-5p}) \times 4] - [c(\text{miR-30d-5p}) \times 1 + c(\text{miR-30a-3p}) \times 3]$ (Supplementary Fig. 16b). This model contains simplified mathematical operations of multiplication, addition, and subtraction that have been successfully validated with DNA IC-CLA as mentioned above, yet can achieve an excellent classification accuracy of AUC over 0.98 in both the training and validation sets. The selected classification model can achieve an NSCLC diagnostic sensitivity of 100%, specificity of 85%, and accuracy of 98% for the validation set. This lays a good foundation for using DNA IC-CLA for the diagnosis of NSCLC.

### Cancer diagnosis with DNA IC-CLA

The extremely low concentrations (<pM) in blood and short similar sequences (18–22 nt) of miRNAs resulted in a big obstacle for applying this DNA IC-CLA for diagnosis in clinical samples. Plus, the relative ratios of different miRNAs are key information for disease classification and, therefore, should not be perturbed during the amplification or sample preparation process. To resolve this issue, we adopted the linear after the exponential PCR (LATE-PCR) to achieve near-linear amplification of multiple miRNAs from low concentrations (≤pM) to a higher detectable range (>nM) without obvious changes of their ratios[43,44]. As shown in Fig. 4, the extracted miRNAs from serum were first hybridized to the stem-loop primers and then reverse-transcribed into cDNAs. Next, the cDNAs were amplified using LATE-PCR by controlling the ratios of two primer pairs (excess and limiting primers). In this step, miRNA can be amplified by LATE-PCR and transformed into ssDNA for subsequent DNA computation without disturbing their original quantity ratios. As shown in Fig. 4c–f and Supplementary Fig. 17, the fluorescence and gel electrophoresis show the linear relationship between the final concentrations of ssDNA products and the initial miRNA concentrations ranging from 1 fM to 1 pM, demonstrating that LATE-PCR can amplify miRNAs and maintain the initial miRNA concentration information. In addition, we used synthetic cDNA and miRNAs to investigate the amplification efficiency of LATE-PCR. Supplementary Fig. 18 shows that the concentrations of ssDNA products were linear to the initial logarithm concentrations of cDNA and miRNA. 1–1000 fM cDNA and miRNA produced 7–24 and 4–20 nM ssDNA products, respectively, after LATE-PCR amplification, which were detectable for DNA IC-CLA. Supplementary Fig. 19 shows a similar amplification efficiency when using 4 different miRNAs as inputs. At this stage, miRNA can be amplified through LATE-PCR and converted into ssDNA for subsequent DNA computations while preserving their original quantity ratios without disruption. These data collectively confirmed the effectiveness of the miRNA amplification process and paved the way for applying DNA IC-CLA for cancer diagnosis in real clinical samples.

To experimentally validate the capability of DNA IC-CLA to distinguish NSCLC and healthy samples, we applied DNA IC-CLA to synthetic samples that match the sample profiles and concentrations

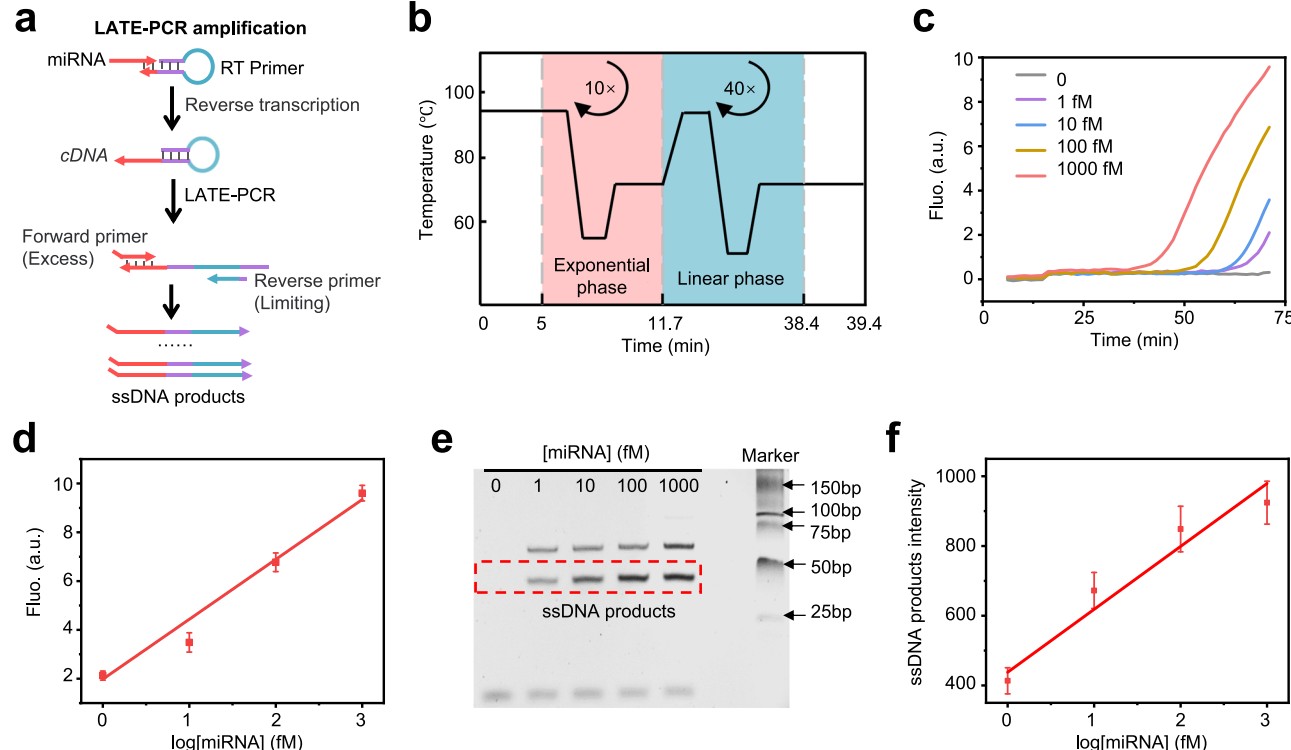

**Fig. 4 | Near linear amplification of miRNAs in clinical samples. a** Schematic illustration of the LATE-PCR amplification for miRNAs. **b** LATE-PCR contains an initial exponential amplification phase and linear amplification phase. The initial exponential phase amplifies dsDNA templates at a higher annealing temperature. The linear amplification phase amplifies the ssDNA products at a lower annealing temperature. **c** Detection of LATE-PCR amplification products with different initial concentrations of miRNA (miR-148a-3p) using a FAM-labeled TaqMan probe. **d** Plot of fluorescence of PCR at cycle 50 versus initial miRNA (miR-148a-

3p) concentrations, demonstrating the linear amplification of LATE-PCR with initial miRNA concentrations from 1 fM to 1 pM. $R^2 = 0.98$. Data are presented as mean values ± SD, $n = 3$ biological replicates. **e** Native PAGE analysis of LATE-PCR amplification products with different initial concentrations of miRNA (miR-148a-3p). **f** Plots of fluorescence intensity of the ssDNA product bands versus initial miRNA (miR-148a-3p) concentrations. $R^2 = 0.95$. Data are presented as mean values ± SD, $n = 3$ biological replicates. Source data are provided as a Source Data file.

calculated from the TCGA database. Only a small subset, ~5.7%, of clinical samples in the TCGA database, are categorized within the challenging-to-classify range (Supplementary Fig. 14c). To enhance the likelihood of successful classification and to validate the performance of the DNA IC-CLA more stringently, we designed and included synthetic samples that are beyond the classification threshold. We mixed 4 synthetic miRNAs at reported ratios (concentrations of femtomolar) to prepare 30 synthetic samples containing 15 NSCLC and 15 healthy samples based on the miRNA-seq data (synthetic sample information is shown in Supplementary Table 1). The replication samples underwent LATE-PCR amplification and then were transferred to the solution that contained DNA IC-CLA. The diagnostic results of these synthetic samples can be obtained by recording the fluorescence signals of FAM and ROX of DNA IC-CLA. As shown in Fig. 5a–d, only 3 NSCLC samples were misdiagnosed as healthy samples, and the remaining 12 NSCLC and 15 healthy samples were correctly classified with a sensitivity of 80.0%, specificity of 100%, and accuracy of 90.0%. To figure out the reasons for the decreased fluorescence differences between FAM and ROX, as well as the decreased accuracy of synthetic miRNA samples compared to the mimetic samples with single RNA (or cDNA) inputs, we compared the LATE-PCR amplification efficiencies of a single miRNA and a mixture that contained other miRNAs in buffer solutions. Our results showed that LATE-PCR amplification of miRNAs could interfere with other miRNAs present simultaneously (Supplementary Fig. 20), which resulted in slightly inconsistent amplification efficiency of different miRNAs and thus might induce concentration errors post-amplification. We believe that a more careful primer design may help relieve this issue. We finally applied DNA IC-CLA for clinical NSCLC diagnosis with serum samples from 25 NSCLC patients and 25 healthy

individuals (sample information is summarized in Supplementary Table 2). The workflow is shown in Fig. 5e. We first extracted total miRNAs in clinical serum samples and amplified the four miRNA markers by LATE-PCR. The amplified samples were added to tubes that contained 5 nM DNA IC-CLA for classification. As shown in Fig. 5f and Supplementary Fig. 21, 19 out of 25 NSCLC samples were correctly diagnosed with a sensitivity of 76.0%, while 20 out of 25 healthy samples were correctly diagnosed with a specificity of 80.0%, and the total accuracy of the DNA IC-CLA for clinical NSCLC diagnosis was 78.0%. Likewise, LATE-PCR amplification of miRNAs may also be interfered with by other miRNAs present in clinical serum samples, which may lead to decreased classification accuracy. Another reason may be ascribed to the concentration discrepancy between the four selected miRNAs in the serum samples and tissue samples from the TCGA database[45,46]. Further large-scale transcript sequencing data in the ongoing project for cancer screening may provide a more accurate miRNA biomarker profile and help resolve this problem.

## Discussion

Single biomarker typically exhibits a non-neglectable individual difference that results in diagnosis bias. Multiple biomarkers can cover a more comprehensive profile and, therefore, enhance the accuracy of cancer diagnosis. DNA computation has the parallel computing capability for multiple inputs, providing a feasible path for realizing automatic and rapid cancer diagnosis. In this study, we have devised a localized DNA classifier based on neuromorphic architecture by integrating the DNA molecular computing core on the surface of the DNA origami framework to analyze complicated multi-biomarker profiles for clinical diagnosis. Our DNA IC-CLA is a strong effort to exploit DNA

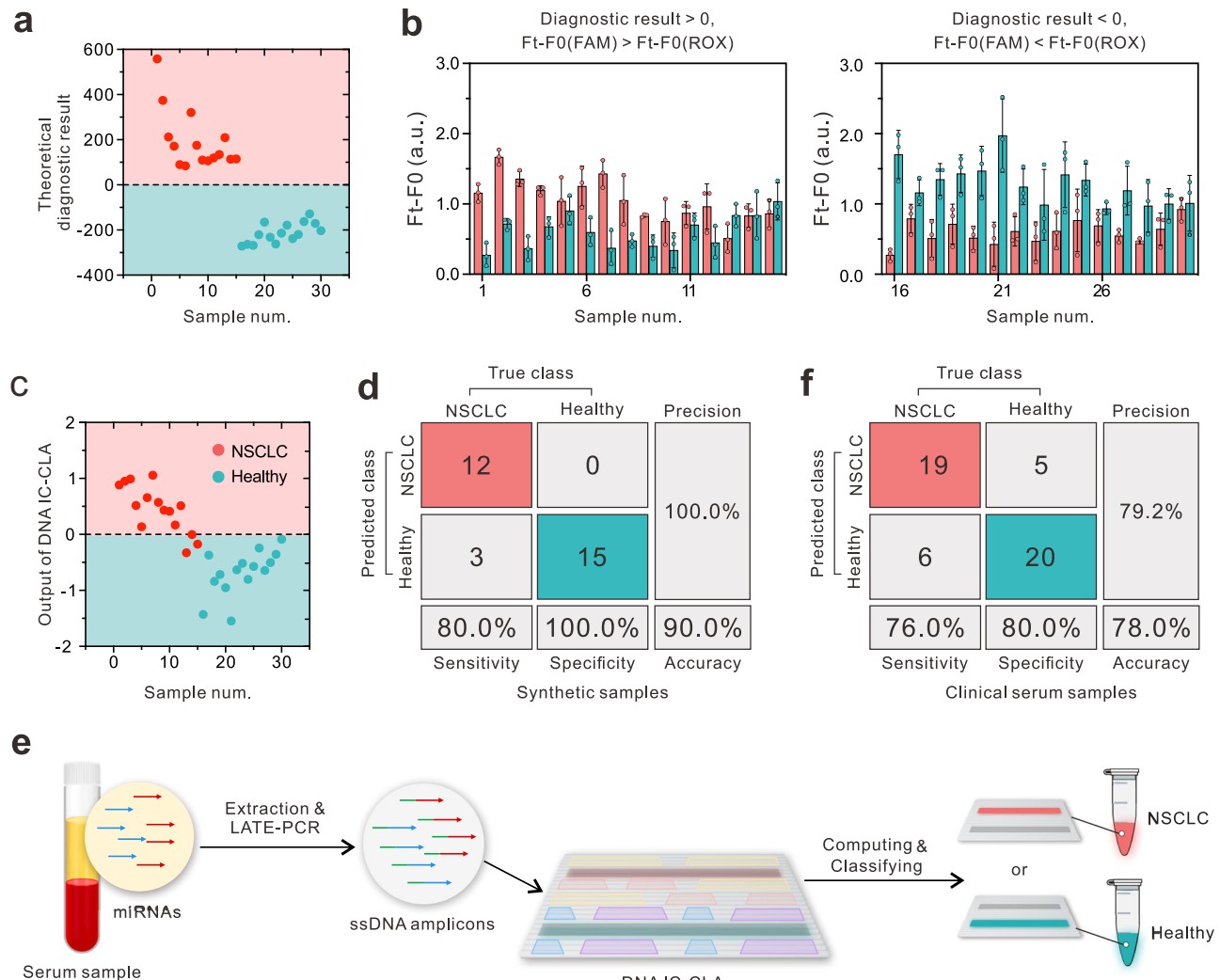

**Fig. 5 | Validation of cancer diagnosis by DNA IC-CLA with synthetic and clinical samples. a** The theoretical diagnostic results of synthetic NSCLC mimicking samples (labeled 1–15) and synthetic healthy samples (labeled 16–30). **b** Fluorescence difference ($F_t$–$F_0$) of the DNA IC-CLA with synthetic NSCLC mimicking samples (labeled 1–15) and synthetic healthy samples (labeled 16–30). In principle, a diagnostic result with a value > 0 or <0 theoretically corresponds to fluorescence signals where [FAM] is greater than [ROX] or [FAM] is smaller than [ROX], respectively. Data are presented as mean values ± SD, $n$ = 3 biological replicates. **c** The classification outcomes of the DNA IC-CLA with 30 synthetic samples. **d** Confusion matrix analysis of the 30 synthetic samples, where a value of $[F_t–F_0(\text{FAM})]–[F_t–F_0(\text{ROX})] > 0.05$ indicates NSCLC, and $[F_t–F_0(\text{ROX})]–[F_t–F_0(\text{FAM})] > 0.05$ indicates a healthy state. **e** Schematic illustrations of NSCLC diagnosis process using DNA IC-CLA with clinical serum samples. **f** Confusion matrix analysis of the 50 clinical samples. Source data are provided as a Source Data file.

computational nanodevices to intelligently sense multiple biomarkers and report cancer diagnosis information based on the neuromorphic architecture. The spatial constraints provided by the DNA origami framework facilitate rapid kinetics in DNA computation and help to reduce interference between probes positioned in distinct zones, particularly those that are further apart. Thus, by enhancing the classification speed of DNA IC-CLA, we can significantly reduce the diagnostic timeframe (Supplementary Table 3 for comparison with other DNA molecular computation systems). Furthermore, reducing interference among various probes will decrease the likelihood of erroneous diagnoses. Both the acceleration of decision-making and the enhancement of diagnostic precision are of paramount importance for clinical applications. Also, our DNA IC-CLA explored integrating automated equipment with molecular computing systems for one-step diagnostics without manual data analysis based on multiple miRNA profiles. We anticipate that the distinctive capabilities of DNA computation, which do not require human intervention for data analysis, will justify future endeavors in developing more robust platforms,

such as DNA molecular computation-based in situ analysis and diagnosis.

The in silico-trained classifier model encoded cancer diagnostic information can be executed by DNA IC-CLA that realized sensitive and accurate classification with clinical samples from cancer and healthy individuals. DNA origami stands out as an ideal candidate for the integration of molecular circuits. In the realm of DNA molecular computing, DNA molecules serve as both inputs and outputs, as well as the building blocks of circuits, making DNA origami particularly advantageous for the precise positioning of probes and circuit elements. Beyond the assembly of integrated molecular circuits, DNA molecules are commonly the analytes in bio-analysis and molecular diagnostics, which makes the development of DNA origami-based molecular circuits even more suitable for downstream diagnostic applications. Although the input numbers of DNA IC-CLA are limited by the finite room on the DNA origami framework, this could potentially be further expanded to include more inputs and sufficient probes with hierarchical self-assembly of larger-sized origami framework for

more complex disease classification. Moreover, the modular design and enhanced physiological stability of structural DNA nanotechnology, which surpasses that of small ssDNA and dsDNA, position DNA IC-CLA as a promising tool for in vivo applications. It holds the potential to perform in situ computations within living cells and even execute actions in response to intracellular biomarkers. This development is poised to inspire a wider range of diagnostic applications.

## Methods

The study was approved by the Ethics Committee at Renji Hospital, School of Medicine, Shanghai Jiao Tong University (protocol number: KY2023-062-B). All methods were performed in accordance with these approved guidelines.

### DNA and RNA sequences and reagents

All DNA and RNA sequences used in the study were purchased from Sangon Biotech (Shanghai, China). The staple strands for DNA origami assembly were obtained and used without further purification (standard desalting). The computing probes and reporting probes of the computing core were purified by high-performance liquid chromatography–purified. The sequence design principle and all sequences of DNA probes are shown in Supplementary Note 4 and Supplementary Table 4. Single-stranded M13mp18 7249 was purchased from Tilibit Nanosystems (Cat. no. M1-12). Other chemicals were purchased from Sigma-Aldrich.

### Preparation of DNA origami framework

The computing probes and reporting probes of the computing core were immobilized on the DNA origami framework by extending the handle sequences (20 nt) to the ends of the corresponding staple strands. DNA origami framework was annealed with 10 nM M13 scaffold (7249 nt) and 5-fold excess of staples in 1× TAE/Mg²⁺ buffer (1× Tris–acetate–EDTA, 10 mM MgCl₂). Annealing was performed in a thermal cycler (Bio-Rad) by incubation at 90 °C for 5 min and then slowly cooling to 20 °C at a rate of 1 °C per 1 min. The DNA origamis were purified by size exclusion chromatography using Sepharose CL-4B (Cytiva GE Life, Cat. no. 17015001) to remove excess staple strands. Sequences of staples of the DNA origami framework are shown in Supplementary Note 5 and Supplementary Table 5.

### Preparation and purification of DNA probes

Computing probes ($L_n$–$N$) were prepared by mixing the backbone $L_n$ and the corresponding number of protruding ssDNA sequences that would be pre-locked on the surface of DNA origami at the same concentration and annealing in a thermal cycler by incubation at 90 °C for 5 min and followed by cooling to 25 °C at a rate of 1 °C per 1 min. The prepared computing probes were subsequently purified by 12% polyacrylamide gel electrophoresis (PAGE); the bands were visualized using ultraviolet light, cut out, and suspended in 1× TAE/Mg²⁺ buffer at 4 °C for 24 h. The resultant solvent was extracted and the concentrations of purified computing probes were determined by NanoDrop (Thermo Fisher).

Two reporting probes (RE and RF) and other DNA duplexes consisted of a bottom strand and a top strand and were prepared by mixing bottom and top strands in a 1:1.2 ratio in 1× TAE/Mg²⁺ buffer and annealing at 90 °C for 5 min, followed by cooling to 25 °C at a rate of 1 °C per 1 min. Excess top strands can help to reduce the leaking reactions. $S$ probes were further purified with 12% PAGE. Finally, these probes of the computing core were kept at 4 °C for future use.

### Preparation of DNA IC-CLA

The computing probes and reporting probes of the computing core were immobilized on the DNA origami framework by mixing the computing probes and reporting probes with purified DNA origami framework (2:1 in molar ratio of computing probes and reporting probes to the docking sites on origami) and kept on a ThermalMixer (Eppendorf) with a mixing frequency of 350 r.p.m. overnight at room temperature. Then, the samples were purified by size exclusion chromatography using Sepharose CL-4B to remove unbound DNA probes. To characterize the DNA origami framework and the DNA IC-CLA, 1% agarose gel in 0.5× TBE/Mg²⁺ buffer (0.5× Tris–borate–EDTA, 10 mM MgCl₂) was run at 70 V for 2 h at room temperature and stained by 4S GelRed. The cropped and uncropped gels are shown in Supplementary Figs. 2a and 22, respectively.

### SVM training and validation

The miRNA-seq data for the SVM classifier training were obtained from the TCGA database. First, we selected 630 NSCLC and 63 healthy samples as training sets to train the linear SVM classifier. We analyzed the differential expression between the two groups and selected 33 up-regulated and 22 down-regulated miRNAs in the NSCLC group to train the classification models. Second, we set some constraints, such as limiting the weights to integers lower than ten and limiting the mathematical operations to summation, multiplication, and subtraction to the 55 miRNA inputs to obtain a minimal set of inputs. Then, we selected the SVM classifier model with the highest AUC value and a reasonable number of miRNA inputs and validated the classifier with 270 NSCLC and 27 healthy samples.

### miRNA extraction in serum samples

All serum samples were collected from Renji Hospital (Shanghai, China) with informed consent and approved by the Ethics Committee at Renji Hospital, School of Medicine, Shanghai Jiao Tong University. Publishing the data from multiple indirect identifiers mentioned in Supplementary Table 2 is approved by the Ethics Committee at Renji Hospital, School of Medicine, Shanghai Jiao Tong University. Total miRNA in serum samples was extracted using the Qiagen miRNeasy Serum/Plasma Kit (Cat. No. 217184) according to the manufacturer's instructions. In brief, a lysis reagent was first added to lyse the serum samples, then chloroform was added to separate the lysate. Subsequently, the samples underwent washing and elution process, and the eluted miRNA was stored in nuclease-free water at −80 °C until needed. The protocol of miRNA extraction is shown in Supplementary Table 6.

### Reverse transcription and LATE-PCR

Synthetic or extracted miRNAs were first reversely transcribed into cDNA by using a reverse transcription kit (Sangon Biotech, Cat. No. B532453) according to the manufacturer's instructions. The conditions and protocol of reverse transcription are shown in Supplementary Table 6. LATE-PCR was performed in PCR buffer containing template cDNA from reverse transcription, excess primer, limiting primer, TaqMan probe, 2× Hotstart PCR Master Mix without dye (Sangon Biotech, Cat. No. B639288). The conditions and protocol for LATE-PCR are shown in Supplementary Table 6.

### Fluorescence kinetic measurements

The arithmetical operations containing multiplication, summation, subtraction, and the entire computation for the classification of DNA IC-CLA were validated by fluorescence kinetics. Fluorescence kinetics experiments were performed in a 96-well plate with 50 μL reaction mixture per well and monitored using a multi-detection microplate reader (BioTek). All reactions were carried out in 1× TAE/Mg²⁺ buffer and kept at 37 °C throughout the reaction. Excitation/Emission wavelengths were set to 492/518 nm for FAM and 585/615 nm for ROX, respectively. For multiplication, reactions were carried out with 5 nM of DNA IC-CLA, 25 nM of each $H$ probe, 25 nM of each $M$ probe, 200 nM of each $M$ reporter, and 20 nM of each Input, respectively. For summation, reactions were carried out with 5 nM of DNA IC-CLA, 25 nM of each $H$ probe, 25 nM of each $M$ probe, 150 nM of each $S$ probe, 200 nM of each $S$ reporter, and 20 nM of each Input, respectively. For

subtraction, reactions were carried out with 5 nM of DNA IC-CLA, 200 nM of $N$ probe, 200 nM of each Fuel probe (Fuel $E$ and Fuel $F$), 200 nM of each reporter, and different concentrations of $E$ and $F$ probes. The conditions of the fluorescence kinetics experiments are shown in Supplementary Table 6.

## Statistics and reproducibility

The statistical analysis was conducted using a two-tailed unequal variance $t$-test for the comparison between two experimental groups. $P < 0.05$ was considered statistically significant. We followed the routine biological replicate requirement in the experiment section, $n = 3$ for each group. All values are presented as means ± SD. For the atomic force microscope, experiments were repeated by at least three times to obtain similar results. Randomization is not relevant to our study.

## Reporting summary

Further information on research design is available in the Nature Portfolio Reporting Summary linked to this article.

## Data availability

All data supporting the results of this study are available within the paper and its Supplementary Information. Source data are provided with this paper. The miRNA-seq data used in this study are available from the TCGA database https://portal.gdc.cancer.gov. Source data are provided with this paper.

## Code availability

The codes of simulations for multiplication, summation, and subtraction are available from GitHub at https://github.com/mzhan167/spatially_localized_DNA_classifier[47]. For SVM training and validation, we adapted a previously published code available at https://www.csie.ntu.edu.tw/~cjlin/libsvm[48].

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

## Acknowledgements

This work was supported by grants from the National Key Research and Development Program of China (2021YFA0910100) to C.Z.; by grants from the National Key Research and Development Program of China (2021YFA0909400), the National Natural Science Foundation of China (22225402, 21974087) and Shanghai Municipal Education Commission-Gaofeng Clinical Medicine Grant Support (2018709) to D.H.; by grants from the National Natural Science Foundation of China (22204099), the Mount Taishan Scholar Young Expert (No. tsqn202312249) and China Postdoctoral Science Foundation (BX2021192, 2022M712103) to L.Y.; by grants from the National Natural Science Foundation of China (22104085) and Shanghai Sailing Program (21YF1424700) to C.Z.; by grants from Shanghai Sailing Program (23YF1451300) to Q.M.

## Author contributions

L.Y. and D.H. conceived and designed the study; L.Y., Q.T., and Q.M. performed the experiments; M.Z. carried out the data simulations; L.Y., Q.T., M.Z., Y.T., X.C., R.X., Q.M., P.G., C.Z., and D.H. supported the optimization of assays and analyzed data; L.Y. and D.H. wrote the manuscript. All authors reviewed the manuscript and approved the final version.

## Competing interests

R.X., Q.M., and C.Z. are employees of Intellinosis Biotech Co., Ltd., with equity in the company. The remaining authors declare no competing interests.
