## [Peer Review File · Nature Communications]

A spatially localized DNA linear classifier for cancer diagnosisReviewer #1 (Remarks to the Author):

See attached document.

Reviewer #1 Attachment on the following page

In this manuscript, Yang et al. report a spatially localized DNA classifier (DNA IC-CLA) that can perform molecular computation for cancer diagnostics. The authors propose that spatially localizing DNA components on a two-dimensional DNA origami enables faster and more effective reactions than in free solution. Da Han and co-workers have demonstrated numerous reaction networks that can process multiple inputs and provide accurate disease diagnostics (e.g. cancer, viral, bacterial infection) [Zhang et al., *Nat. Nanotech.* 2020; Ma et al., *Sci. Adv.* 2022]. The research is presented thoroughly and neatly, and the results are promising. However, it is difficult to realize the novelty of this study compared to the previous classifiers. Spatially localizing DNA components seems to improve the speed of the entire computation, but only by approximately two folds in 180 minutes. This might have been achieved by simply increasing the concentration of the probes in, for example, the multiplication process. Moreover, the distance dependency of molecular computation is not thoroughly investigated in this paper. Overall, I do not think this paper is suitable for publication under *Nature Communications*.

1. What is a neuromorphic architecture in molecular computation? The authors describe such capability mostly in the abstract, introduction, and conclusion, but hardly in their results. The authors could consider each node in their classifier an example of a neuromorphic structure, but how is such claim different from their previous studies?
2. As shown in Figure 1c and Supplemental Figure 1, are there specific reasons behind the positions of the computing zone (specifically for the different probes), as well as the reporting zone? How are the authors able to confirm that the computing probes and reporting probes are positioned in the origami as designed?
3. The differences in molecular computation between a system in which components are localized and that in free solution are not clearly explained. Other than the idea that the components become more available in a defined space, what may improve the processivity of such reactions? Perhaps a simulation that contrasts the molecular interactions between the two conditions would be helpful here.
4. In Figures 2b,d, and f, how were the authors able to measure the fluorescence of the probe localized to the surface of DNA IC-CLA? I assume that the fluorescence kinetics and endpoint measurements were conducted using the microplate reader. The authors should clarify the conditions used to track the fluorophore-modified strands attached to the origami.
5. Is Supplementary Fig. 6 a simulation of the subtraction? Lines 174 to 176 in page 6 seems to imply this, but the parameters and equations of the simulation are unclear. Moreover, were simulations conducted considering spatial localization of probes and reporters?
6. As shown in Figure 3a, why is it specifically during subtraction that DNA IC-CLA demonstrates a much faster kinetics than in free solution? If the kinetics of multiplication and summation are not too different for the two cases, it seems like only the subtraction components need to be localized?

7. The term reaction rate in Figure 3 is not very clear. Is this based on the slope of the fluorescence kinetics? If these results were to be simulated, would the reaction network exhibit different strand displacement rate constants when localized? The authors should also rationalize why they only measured the reaction rate for the first ten minutes. Why not measure the reaction rate of the entire reaction?
8. Figure 3c does not appear to be a fluorescence kinetics plot, but rather an endpoint measurement of fluorescence. Nonetheless, Supplementary Figure 7 shows that the background signal increases for both localized and non-localized kinetics. Therefore, claiming that the localized components are more robust, does not seem very convincing.
9. On page 5 line 113, the authors mention possible crosstalk between the probes and the incorrect targets. However, crosstalk would most likely be caused by the complementarity of sequences and domains, so this may occur for the spatialized probes as well. In addition, if crosstalk is an issue, this may be visualized as a substantial fluorescent signal leak, but Supplemental Figure 8 shows minimal differences in background signal between the localized and non-localized kinetics. It is unclear what the authors imply by crosstalk and the potential issues that result from such crosstalk of probes and targets.
10. The authors on page 5 line 117 claim their design as “modular”, but it is unclear what aspect of the reaction network can be considered modular. My understanding of modularity is being able to swap out different sequences or domains of probes and reporters onto the origami surface, and yielding similar outputs. The authors should discuss this claim thoroughly based on their results.
11. Supplementary Table 4 does not provide any information regarding the conditions of the fluorescence kinetics. Under the Methods section, the authors should specify the concentrations, measurement protocols, and other details regarding the use of the microplate reader.
12. The conclusions section currently describes the general advantages of a molecular classifier for diagnostics. The authors should elaborate on the importance of speed and minimizing crosstalk for DNA IC-CLA. What other implications can such improvements in speed and accuracy provide?

Reviewer #2 (Remarks to the Author):

"I co-reviewed this manuscript with one of the reviewers who provided the listed reports. This is part of the Nature Communications initiative to facilitate training in peer review and to provide appropriate recognition for Early Career Researchers who co-review manuscripts."

Reviewer #3 (Remarks to the Author):

This article reported a DNA molecular diagnostic system that is spatially positioned and integrated on DNA origami and is based on the credible SVM linear classifier as the calculation principle. It used multiple miRNA inputs as detection targets, and ultimately achieved faster and more effective cancer diagnosis than traditional free diffusion DNA diagnostic systems. I think this article provides a good idea for the design of future molecular diagnostic systems, which is of great significance at a time when disease diagnosis is required to be faster, more accurate, more intelligent, and more integrated. I recommend publishing after solving the following questions:

1. In page 3, line 76-77, the abbreviation rule of the acronym DNA IC-CLA should be explained by adding underline below specific letters.
2. I can understand the DNA molecular diagnostic system is based on a SVM linear classifier, but the "neuromorphic architecture" claimed by the authors is not comprehensive. It is recommended to reconsider the usage of the word "neuromorphic" or give clear description about the principle.
3. In Figure 2b and 2d, the concentration of input a, b, c, and d should be noted in the figure legend.
4. In Page 10, line 277-278, "b, Fluorescence kinetics of the multiplication process of non-localized system", fluorescence kinetics data were not found in Figure 3b.
5. The author compared the multiplication computation of non-localized system and DNA IC-CLA under different preparation time in Fig 3b and compared the entire computation in Fig S7. However, only slight difference was observed when comparing the entire computation. Therefore, it is not convincing to claim that the origami-integrated strategy exhibits higher robustness because comparison result of the entire computation is more important for the system. Besides, I wonder why the difference is more obvious in multiplication computation?
6. In Figure 4d, 4f, S13b, S14c, the author plotted the fluorescence of PCR at cycle 50 or band intensity versus initial miRNA concentrations but used logarithms in horizontal axis. In my opinion, here using logarithms means that the amplification of LATE-PCR is not linear. Then how to avoid the nonlinear deviation caused by such an amplification when performing SVM classification? Besides, names and sequences of target miRNAs used in Figure 4, Figure S13 and Figure S14 should be noted in Figure legend or SI. Error bars should be added in Figure 4 and S13.
7. In line 383, why using $[Ft-F0(FAM)] - [Ft-F0(ROX)] > 0.05$ and $[Ft-F0(ROX)] - [Ft-384 F0(FAM)] > 0.05$ instead of $[Ft-F0(FAM)] - [Ft-F0(ROX)] > 0$ and $[Ft-F0(ROX)] - [Ft-384 F0(FAM)] > 0$? The reason choosing 0.05 should be explained. Besides, what is the proportion of samples entering the range with $[Ft-F0(FAM)] - [Ft-F0(ROX)] < 0.05$ or $[Ft-F0(ROX)] - [Ft-384 F0(FAM)] < 0.05$?
8. Units and column names should be added to the table in Figure 5a.
- 9: The conclusion mentions that the DNA IC-CLA can be reused, but there seems to be no experiment to prove this in the text and SI. Besides, the author also claimed that the design of this work has higher biostability (line 404 of the text), but there is also no experiment to prove this in the text and SI. Please correct here or add corresponding experiments.
10. In Figure S2d, the horizontal axis is displayed incompletely.
11. In line 243 and Figure S8, what is the calibration system? Explanations and discussions should

be added.

12. In Figure S8, there seems to be no difference in the fluorescence kinetics between non-localized system and DNA IC-CLA seems? In the cx1 group, the non-localized system even showed a slightly higher kinetics, which is conflicted with the discussion in line 215-218 of the main text. It is recommended to test three or more parallel samples to characterize the kinetics.

13. In Figure S9, different combinations of DNA input concentrations should be given in the figure legend.

14. In Figure S10, it is better to label streptavidin in both red and blue reporting zones with different patterns. What is difference between the 16 imaged origami in the bottom? Different combinations of DNA input? It should be noted in the legend.

15. In Figure S11. This result is not convincing. It is recommended to reconstruct the origami imaging or just remove this figure.

16. In Figure S12, the legend is too concise to be understood. What is FRD? What is the meaning of fold change? Explanations should be given.

17. In line 327 and Figure S15, what is the concentration of the miRNAs amplification products after LATE-PCR amplification?

18. In Figure S16e, it is better to use percentage as the unit for the vertical axis.

19. In Table S4, many characters are not displayed correctly. Please check the table careful and correct the errors.

20. In Table S4, in the fluorescence kinetic experiment condition, why choosing 37°C instead of room temperature? Are there any considerations?

Reviewer #4 (Remarks to the Author):

Reviewer #5 (Remarks to the Author):

General Comments

This work presents a localized molecular classifier where DNA strand displacement computations are localized on a DNA origami. The authors present each of the computing blocks separately. They show how this localized classifier is faster than having just the same computing elements in bulk. Finally they show how this classifier, combined with a LATE-PCR amplification can be used in a clinical diagnosis setup.

Globally the work is impressive because it combines 3 different expertises of the DNA nanotechnology field: structural DNA nanotechnology, DNA computing and medical diagnostics. These thematics resonate with a multitude of other fields: material science, computer science, physics, biochemistry and biology. Therefore this reviewer believes it will be of interest for the broad audience of Nature communications, once the authors address several crucial points.

Addressing those points is necessary to properly assess this work and put it in perspective with other existing techniques as well as future developments in the field.

First, one of my major concerns is that it is hard to understand the whole scheme and how the modules are arranged together. The resolution of the figures is low, and the labels of the DNA domains are barely legible. The captions of the supplementary figures are terse (usually one nondescript sentence that does not do justice to the figure). The schemes are hard to understand because it is not clear what counts as a reagent and what counts as a product (for instance they write chains of reactions like $A+B \rightarrow C+D \rightarrow E+F$. In this case, is D a reagent for the second reaction or a product of the first reaction? It is unclear. Chemists have conventions for this. In the chain above, C and D would be product of the first reaction, and any additional reagent for the second reagent would be indicated by a curved arrow merging with the main arrow of the reaction). Another related concern is that the method section does not clearly reflect what was done. There is a lot of emphasis on the preparation of the components (assembly, purification...), but little on how the computation was actually done experimentally. Also the authors do not explain how sequences were designed, although the homology of domains seen in strand displacement typically places strong constraints on design.

Another concern is that the authors tend to emphasize positive results (like correct classification), while avoiding discussing negative results (like the severe background seen in some fluorescence experiments). They should discuss and experimentally characterize this, rather than putting it under the rug. For instance, due to the nature of strand displacement, some strands are likely to bear complementary domains and fleetingly bind, which may cause leakage. For instance the helper strand can bind to the M probes to form a complex that leaves a toehold open for toehold exchange (This is easy to check on Nupack). The authors should better discuss and experimentally measure this.

In my mind the authors need to

- Assess precisely Performance and Speed of classifier (Experiment needed)
- More quantitative and precisely compare themselves with state of the art for speed and performance
- Estimate (orders of magnitude) how/why localized DNA computing is an improvement over other methods.
- Present more clearly and fairly input datasets used for classification
- Add information to figure Legends in the supplementary material
- Correct typos in the schemes
- Add high resolution images

After having addressed those concerns, I believe that the paper will be suitable for publication in Nature Communication.

Main questions:

*Design/testing/optimization of modules

The authors did not do a good job of showing step by step how the modules were constructed. I would expect them to do experiments showing step by step how the components interact. Due to the nature of strand displacement, all domains must be present in solution from the start. This creates a potential for auxiliary strands to interact in absence of any input. For instance the helper strand H_a will bind to the M-probe (which can be verified in Nupack), creating a potential for leak. This may explain the strong fluorescent background observed in Figure 2 (b and d). I would like to see experiments that better investigate this leak. They authors could for instance test what happens to the leak in absence/presence of M probes and in absence/presence of Helper strands for the a input. This would help to pinpoint the origin of the fluorescence background.

Also the authors do not do a good job of showing the linearity of their module. The operation $c(\text{input}) \times \text{Weight}$ which associates a weight to an input concentration is bilinear. It depends linearly both on the input concentration and the weight. The authors did test the linearity for the weights, but they did not test the linearity for the input concentration. I expect them to test that by doing a titration of the input between 0 nM and say 50 nM. I also expect that linearity will break down at low concentration because of the strong fluorescent background. The authors should plot the final raw fluorescence against the concentration of the titrated input, with a fine graining of the

concentrations.

***Clarity of scheme**

The authors should clarify their mechanistic schemes. It is hard to understand what is going on in Figure 2a,c,e. I presume that the E and F probes are released in solution in Fig2c (but it is not clear in Figure 1c if the E probe is a product or reagent). But then why are the E and F probes in a different form in Fig2e (they are partially duplexed). And in Figure 2e, I do not understand why the E probe was released by the origami. It was already released in the precedent step. Also what does the diode symbol mean here? And I am confused about the subtraction scheme. Does it happen in solution or on the origami?

***Why use molecular calculation?**

For detection in pM/fM range the authors need to do a PCR. Then what are the advantages of doing a PCR and molecular calculations, versus doing a PCR and analyzing this on a computer? Is the gain in speed? or in sensitivity? specificity? Please comment in the text

***Quantitative evaluation of Classifier Speed and Performance**

The authors should better benchmark their performance. The performance of a classifier can be defined as the concentration of the closest inputs that can be discriminated into different classes. This is known as the separation margin of the classifier. There is no experiment in the paper that clearly measures the separation margin of their classifier. Figure 2f is a step toward this, but it is not the full classifier, and it does not show the margin clearly.

=> This reviewer expects an additional experiment, with a carefully designed combination of inputs to assess the discrimination limit. For instance, having closer and closer inputs being discriminated into different classes until this is no longer possible.

***Speed** What is the real time saving vs. other bulk molecular methods? How will the authors method scale-up for more complex calculations? Will it scale-up favorably if layers are added / tiles glued together?

This reviewer is curious to see what the trade-off is between performance and speed for this work, and how it compares to other DNA classifiers in bulk. The author should be careful in their comparison. Some papers use a digital subtraction, and some authors (including those of the paper here) use a pre amplification step like LATE-PCR. This reviewer expects a table or a graph (whichever the authors believe more appropriate) comparing this work performance and speed to other molecular classifiers

***Why use DNA origami?**

If the big difference of this paper is bulk vs. localization, then

Do we really need DNA origami?

What does origami really offer compared to strands on a surface or a ball?

What's the point of very fine distance control?

Why put all weights on the same origami?

How can DNA origami formation/reconfiguration be used for computation in the future?

Localizing computation is a different modality from bulk. It deserves to be compared not only with respect to the current implementation but from a more general point of view by extracting the parameters responsible for the different time scales, performances to see the physical limitations and scaling of Bulk vs. localized on DNA origami.

*** What are the limitations of the classification method? How can you go about overcoming them?**

The classification of biological samples is complex because of additional purification steps, but the authors need to be precise about the real difficulty of the classification part:

How close are the inputs that should give different outputs?

What is the relative difference between these 2 close but different classified inputs?

The tables of 4D miRNA concentrations can be replaced by a matrix of relative distance or 2 2D plots showing the distribution of miRNA in the samples. This will allow the reader to have a better grasp of the input data.

This reviewer did such plots for Figure 5. It shows that the data chosen can be classified using a single miRNA marker, and it is unclear what advantage is brought by multi-input classification in this case. The authors chose to use only 30 samples from the 50 appearing in the supplementary materials. How was this choice made? This reviewer believes that showing the limits of the classifier is necessary to really be able to judge its performance.

Having samples that are too close from the frontier to be classified would make the classification task more credible. The current 100% precision on synthetic samples is not fully convincing and seems to depend more on the input dataset choice than the real classifier performance.

=> The authors should present more clearly the input datasets and how hard to classify they really are.

Comments on Figures

Fig1

b: Clear, but Could add quantitative info/table of Performance & Speed of Classifiers in Bulk vs Localized.

c: DNA IC-CLA is not a clear acronym and should be defined also in caption. Would add "DNA origami" keyword for people from a broader audience.

d: The color of the stripes on the breadboard suddenly change after the summation probes are added. Why? Also the anchored strand have changed (coloured anchor before, blackish anchor after. Why?

Fig2

a,c: At first glance, strand displacement is not obvious (it could be a polymerization from a template). Showing displaced strands or adding some symbol might help the reader.

e: "catalytic entropy-driven amplification" does not appear clearly in the figure. The reader is referred to Supplementary Fig 5, but the caption in this figure is terse (only one sentence). More details on this step seem crucial.

Fig 3:

a: Great and clear

b: What is tested is not clear from the figure only. (need to read the caption).

c: What is tested is not clear from the figure only. (need to read the caption) Would add "Classification results for varying inputs" or something similar. Table representation is not adequate: Hard to grasp quickly the composition of each sample. Please choose a more graphical representation and put the table with raw datas in SUPMAT.

Fig. 4:

a: not clear from the figure only. Add title like "LATE PCR amplification"

b: Time in X axis is missing. This is important when assessing the speed of the diagnosis.

c,d,e,f: Great

Fig. 5:

a: The raw data table does not allow us to grasp how the input dataset is organized. Please put the raw data in Supplementary material and add a graphical representation as discussed before (this could be several 2D plots, a distance matrix etc).

b: "Results" in the X axis is not clear.

c-e: Great

d: Great but shouldn't this be the first panel before the current "a"?

Figure S3:

The complementarity of the helper strands seem wrong and all over the place. For input a, a2* strand is listed. For input b, strands b2, b3 and b4* are listed. For input d, strands d2 and d3* are listed (but the low resolution is frankly painful to read, even at maximum magnification). It seems to me that the helper strands Ha should have a domain a2 (not a2*) to help displace the long top strand. Same comment for input b and c. Please correct as needed and make those symbols readable.

Figure S5:

The caption should better describe the figure. I do not understand the first reaction: why are there 4 strands anchored on the origami after the reaction cycle while there was only one before. The new strands do not seem to come from the catalysis cycle since none of them bear a red domain. Overall the scheme is hard to understand (not least because the domains are not legible

Comments on text:

L26: "in a faster and more effective manner" => Be more precise

L62: "Slow kinetics" => How slow ?

L68: "ideal scaffolds" => Why "ideal" ?

L77: "spatially localized DNA classifier (DNA IC-CLA)" => I don't understand the choice of Acronym DNA IC-CLA.

L163: "It is typically difficult to implement negative weights or weighted sums for DNA-based computation" => Please add relevant references

L170: "catalytic entropy driven amplification" => Give a short explanation in main text and add more details in the supplementary materials

L240: "A successful linear classifier requires cascaded and accurate mathematical operations for effective data classification" => What is hard in such cascading ?

Reviewer #6 (Remarks to the Author):

Response to Reviewers' Comments

Reviewers 2, 4 and 6

"I co-reviewed this manuscript with one of the reviewers who provided the listed reports. This is part of the Nature Communications initiative to facilitate training in peer review and to provide appropriate recognition for Early Career Researchers who co-review manuscripts."

Response: we thank the reviewers for providing valuable comments to our work. We have thoroughly revised our manuscript to clearly address the comments in the listed reports.

Reviewer 1:

In this manuscript, Yang et al. report a spatially localized DNA classifier (DNA IC-CLA) that can perform molecular computation for cancer diagnostics. The authors propose that spatially localizing DNA components on a two-dimensional DNA origami enables faster and more effective reactions than in free solution. Da Han and co-workers have demonstrated numerous reaction networks that can process multiple inputs and provide accurate disease diagnostics (e.g. cancer, viral, bacterial infection) [Zhang et al., Nat. Nanotech. 2020; Ma et al., Sci. Adv. 2022]. The research is presented thoroughly and neatly, and the results are promising. However, it is difficult to realize the novelty of this study compared to the previous classifiers. Spatially localizing DNA components seems to improve the speed of the entire computation, but only by approximately two folds in 180 minutes. This might have been achieved by simply increasing the concentration of the probes in, for example, the multiplication process. Moreover, the distance dependency of molecular computation is not thoroughly investigated in this paper. Overall, I do not think this paper is suitable for publication under Nature Communications.

Response: We are grateful to the reviewer for taking time to evaluate our work. However, we would like to respectfully clarify the novelty of this work. In this manuscript, we mainly improve the techniques of molecular computing from the three aspects listed below towards approaching the goals of better classification robustness, computation speed, as well as system complexity.

1. Until now, the intelligence of reported localized DNA devices has been predominantly based on Boolean structures, which respond only to the presence or absence of target molecules. We have introduced a novel localized molecular classifier capable of performing arithmetic operations, such as multiplication, addition, and subtraction, on complex input patterns with varying target concentrations in biological samples, resulting in accurate classification outcomes.
2. Current DNA-based classifiers depend solely on interactions between diffusible molecular components, which are constrained by slow kinetics and the intricate design of probes with high sequence specificity for orthogonal reactions. While increasing reactant concentrations can enhance kinetics, leaky reactions are unavoidable, leading to false positive data. These challenges significantly complicate the design of larger circuits and compromise the error correction capabilities of classifiers. Our DNA IC-CLA demonstrates superior performance over diffusible systems in both computational speed (over 3 times faster) and classification robustness.
3. We have devised a comprehensive scheme for this all-in-one localized DNA classifier,

designed for cancer diagnosis in clinical samples with high sensitivity and specificity. This advancement propels the field of DNA computation towards applications in cell-based biocomputing and medical diagnostics.

Regarding the distance dependency in molecular computation, we have set the distance between each localized probe at 10.88 nm without further optimization. This choice ensures maximum modification density with an adequate number of probes for rapid computation and sensitive detection. Additionally, previous research by Seelig's group (*Nature Nanotechnology*, 2017, 12, 920-927) has demonstrated the superiority of probe modification density around 9-10 nm for enhancing computational speed. Therefore, we are of the opinion that it is not necessary to carry out detailed investigations on the distance dependency of DNA IC-CLA in this work.

Comment 1: What is a neuromorphic architecture in molecular computation? The authors describe such capability mostly in the abstract, introduction, and conclusion, but hardly in their results. The authors could consider each node in their classifier an example of a neuromorphic structure, but how is such claim different from their previous studies?

Response:

A neuromorphic architecture in molecular computation aspires to emulate human cognitive tasks with efficiency and parallelism, leveraging molecular components such as nanoscale transistors, memristors, and other nanoscale devices to construct circuits that mimic the connectivity and functionality of neural networks. Distinct from prior DNA computing devices, which primarily rely on Boolean structures, our DNA Integrated Circuit Classifier (DNA IC-CLA) is envisioned as a neuromorphic architecture integrated into a physical chip. This system features several artificial probes, analogous to neurons, each with assigned positions and predetermined weights and threshold values, enabling it to execute arithmetic operations—such as multiplication, addition, and subtraction—for classification purposes. This concept parallels the groundbreaking work published in *Nature* (*Nature* 2022, 610, 496–501), which showed a proof-of-principle example of neuromorphic architectures for non-linear classification. Therefore, the result of neuromorphic computation is mainly exhibited as a concept and classification results for diagnostics (Yes/No) instead of detailed mapping each component in the system to the corresponding architecture. We have added some explanations of the DNA IC-CLA's neuromorphic architecture in the results section on page 9.

Comment 2: As shown in Figure 1c and Supplemental Figure 1, are there specific reasons behind the positions of the computing zone (specifically for the different probes), as well as the reporting zone? How are the authors able to confirm that the computing probes and reporting probes are positioned in the origami as designed?

Response: The primary design principles for positioning computational and reporting zones involve ensuring that functional areas are segregated to prevent cross-talk and to provide ample space for reporting with a maximum number of probes. To validate the successful delineation of these zones, we employed fluorescence techniques sensitive to the slightest changes of probe modifications, an established method also utilized in other studies (e.g., *ACS Synthetic Biology*, 2021, 10, 2878-2885; *Science*, 2011, 332, 1196-1201), along with atomic force microscopy (AFM) imaging (Fig. S2,

S15) for verification. For instance, as depicted in Figure 2, the fluorescence intensity of each weighting operation aligned well with the intended weight, confirming that the computational probes are precisely localized on the DNA origami as per design. We have included a concise explanation in Supplemental Figure 1 to clearly delineate the design of the computational and reporting zones.

Comment 3: The differences in molecular computation between a system in which components are localized and that in free solution are not clearly explained. Other than the idea that the components become more available in a defined space, what may improve the processivity of such reactions? Perhaps a simulation that contrasts the molecular interactions between the two conditions would be helpful here.

Response: We thank the reviewer for this constructive comment. To illustrate the acceleration of computation by localizing computing probes on our DNA-IC CLA, we have performed a simulation to show a comparison of computation efficiency between non-localized and localized conditions. All kinetic simulations were conducted with Python's SciPy package (version 1.2.1) using the solve_ivp() function.

In the simulation model, we assume that most of the DNA strand displacement reactions are reversible. We consider that branch migration is fast so that the reaction rate will mostly depend on toehold binding and subsequent toehold unbinding, which are represented as a simple bimolecular reaction. Since the reaction rate does not primarily depend on sequence length or base composition for 10-100 nt reactants, we assume that the hybridization rate (k_f) of two single-stranded DNA with complementary sequences is close to $3 \times 10^{-3} \text{ nM}^{-1} \text{ s}^{-1}$. The dissociation rate (k_r) can be determined using $\Delta G^\circ = -RT \ln K$, where $K = \frac{k_f}{k_r}$. Moreover, we set a factor L which denotes the local concentration and captures the effect of localization.

In multiplication step, to simplify the reaction of signal initiation, all the probes that locate on the computing platform are considered as a whole reactant, and helpers that take part in the same multiplication reaction are treated as a reactant as well. Therefore, the reaction of signal initiation can be described as follows:

where w represents the weight we gave to the input.

The following pivotal reactions in multiplication can be described as follows:

We then used I to represent Input, H to represent Helper, W to represent Weight, $actW$ to represent activated Weight, S to represent S probe, $actS$ to represent activated S probe and O

represent the platform. We designed one platform contains 4 repeated functional regions, and therefore we can obtain:

$$\frac{\partial[W]}{\partial t} = k_1 \times [I] \times [H] \times (4 \times w \times [O]) - L \times k_2 \times \frac{[W]}{4 \times w} \times \frac{[M]}{w} + L \times k_3 \times \frac{[actW]}{4 \times w} \times \frac{[W2]}{w}$$

Since this local concentration is difficult to compute in the general case (*Journal of the American Chemical Society*, 2011, 133, 2177-2182), we obtained the value of L by fitting the experimental data and found $L=1$. For summation step, we combined two different multiplication steps with different values of w and make a summation implementation. The five rate constants used in multiplication model are listed below:

Rate constant in multiplication model	Value ($\text{nM}^{-1}\text{s}^{-1}$)
k_1	5×10^{-4}
k_2	5×10^{-4}
k_3	3×10^{-11}
k_4	5×10^{-4}
k_5	3×10^{-11}

The resulting reactions for the subtraction step are as follows: First, E and F probe would bind to the N probe and start a winner-take-all reaction, resulting in excess E or F probes interact with RE probe or RF probe. The rate constants of these reversible interactions that occur via the toehold can be estimated based on toehold length (*Journal of the American Chemical Society*, 2009, 131, 17303-17314). Then, a Fuel strand (Fuel E or Fuel F) can bind to the exposed domain of the waste and restore E (or F) probe. The resulting free E (or F) probe can then undergo a localized-like reaction, in which the restored E (or F) probe interacts with the most adjacent RE (or RF) probe localized on the platform. To simplify the reaction among E probe, F probe and N probe, we assumed them as a one-step reaction. The reactions in subtraction model can be described as follows:

The rate constants used in subtraction model are listed as follows:

Rate constants in subtraction model	Value ($\text{nM}^{-1}\text{s}^{-1}$)
k_6	5×10^{-4}
k_7	5×10^{-11}
k_8	1×10^{-5}
k_9	5.55×10^{-15}
k_{10}	5×10^{-4}
k_{11}	5.97×10^{-16}
k_{12}	1×10^{-5}
k_{13}	8.19×10^{-15}
k_{14}	5×10^{-4}
k_{15}	3.74×10^{-14}

Additionally, we determined the value of L through a fit to the experimental data (Fig. S9 and S10c). This analysis revealed that the interaction rate between E (or F) probe and RE (or RF) probe is L ($L=10$) times more rapid when localized compared to non-localized conditions. In our simulations, we utilized the experimental concentrations of RE (or RF) probes under both non-localized and localized conditions, with the sole variable being L , which was set to 1 for non-localized and 10 for localized conditions. Ultimately, a Reporter complex can bind to the exposed domain of the opened RE (or RF) probe (designated as Re or Rf probe) to elicit a response.

In our simulations, we observed that the model is sensitive to the rate constants of E (or F) binding to RE (or RF) as well as the restoration of E and F probes. This suggests that localization could significantly influence the catalytic step. We have included the methodology and results of these simulations in Supplementary Note 1 and Fig. S9, S10.

Comment 4: In Figures 2b, d, and f, how were the authors able to measure the fluorescence of the probe localized to the surface of DNA IC-CLA? I assume that the fluorescence kinetics and endpoint measurements were conducted using the microplate reader. The authors should clarify the conditions used to track the fluorophore-modified strands attached to the origami.

Response: We appreciate the reviewer's feedback. Fluorescence kinetics experiments were conducted in 96-well plates, with each well containing 50 μL of reaction mixture and monitored at 37°C using a multi-detection microplate reader (BioTek). The reactions were conducted in 1 \times TAE/Mg $^{2+}$ buffer and maintained at 37°C throughout the process. The excitation and emission wavelengths were set at 492/518 nm for FAM and 585/615 nm for ROX, respectively. In response to the comment, we have included a comprehensive description of the fluorescence kinetics and endpoint measurements in the Methods section and Supplementary Table 5.

Comment 5: Is Supplementary Fig. 6 a simulation of the subtraction? Lines 174 to 176 in page 6 seems to imply this, but the parameters and equations of the simulation are unclear. Moreover, were simulations conducted considering spatial localization of probes and reporters?

Response: The previous Fig. S6 was an experimental data of the subtraction. In this revision, we have supplemented a simulation study in comparison with the experimental data. The parameters

and equations of the simulation are stated in Supplementary Note 1 and results of simulation are shown in Figs. S9. The simulation considered spatial localization of probes and reporters as detailed in our response to Comment 3.

Comment 6: As shown in Figure 3a, why is it specifically during subtraction that DNA IC-CLA demonstrates a much faster kinetics than in free solution? If the kinetics of multiplication and summation are not too different for the two cases, it seems like only the subtraction components need to be localized?

Response: In accordance with our design, the multiplication and summation operations are executed through three-step and four-step DNA strand displacement reactions (SDRs), respectively. Typically, these processes do not exhibit substantial kinetic differences (1-1.5 times faster) between localized and solution states, as observed in both experimental and simulation data. However, since subtraction is a five-step SDR cascade that follows multiplication and summation, and is further augmented by entropy-driven catalytic cycles, the kinetic differences between localized and non-localized conditions in subtraction are more pronounced (3 times), as evidenced by our newly incorporated simulation data (Fig. S10). Generally, the more steps involved in SDR, the more pronounced the reaction speed changes we can achieve with the localized system. Moreover, simply localizing the subtraction reaction is not feasible in the case, as other diffusible components would struggle to react as intended due to the complex environment, such as cells and clinical samples. In this manuscript, we aim to develop an integrated, all-in-one localized DNA chip-based classifier for cancer diagnosis in clinical samples, which would also propel the entire field of DNA computation towards applications in cell-based biocomputing (i.e. the whole systems can be delivered to cells together).

Comment 7: The term reaction rate in Figure 3 is not very clear. Is this based on the slope of the fluorescence kinetics? If these results were to be simulated, would the reaction network exhibit different strand displacement rate constants when localized? The authors should also rationalize why they only measured the reaction rate for the first ten minutes. Why not measure the reaction rate of the entire reaction?

Response: We appreciate the reviewer's observation. The reaction rate depicted in Figure 3 is calculated as the slope of the fluorescence kinetics over the initial 10 minutes. This approach is adopted because the reaction network yields products at a rate that is approximately linear during the first 10 minutes, which is more reflective of the actual reaction rate for DNA computing cascades. As the reaction progresses, the rate gradually decreases due to changes in the concentration of each probe. This clarification has been incorporated into the revised manuscript (Page 8).

Comment 8: Figure 3c does not appear to be a fluorescence kinetics plot, but rather an endpoint measurement of fluorescence. Nonetheless, Fig. S7 shows that the background signal increases for both localized and non-localized kinetics. Therefore, claiming that the localized components are more robust, does not seem very convincing.

Response: We appreciate the reviewer's comment. Figure 3c represents an endpoint measurement of fluorescence rather than a fluorescence kinetics plot, and we have updated the

legend accordingly. The previous Fig. S7 illustrates that both localized and non-localized kinetics exhibit background signal increases, which are due to unavoidable random errors resulting from equipment and/or sample processing. To account for this background effect, it is more appropriate to assess the robustness of the two systems using the change in fluorescence ($F_t - F_0$). The $F_t - F_0$ remained consistent after one week for the localized DNA IC-CLA, whereas it significantly decreased for the non-localized system, indicating that localized components are more robust. The bar graph statistics have been revised and are now presented in Fig. S11.

Comment 9: On page 5 line 113, the authors mention possible crosstalk between the probes and the incorrect targets. However, crosstalk would most likely be caused by the complementarity of sequences and domains, so this may occur for the spatialized probes as well. In addition, if crosstalk is an issue, this may be visualized as a substantial fluorescent signal leak, but Supplemental Figure 8 shows minimal differences in background signal between the localized and non-localized kinetics. It is unclear what the authors imply by crosstalk and the potential issues that result from such crosstalk of probes and targets.

Response: We appreciate the reviewer's observation. Our intention was to convey that the DNA IC-CLA can mitigate leakage by purifying the localized probes in conjunction with the DNA Origami structure and by minimizing undesirable physical interactions between different types of probes. This is achieved by confining them within distinct computing zones that are sufficiently separated. However, the similarity in background signal between localized and non-localized components might be attributed to the spatial proximity of probes within each computing zone and/or an increased local concentration. To prevent any confusion, we have refrained from using the term "crosstalk" in the revised manuscript.

Comment 10: The authors on page 5 line 117 claim their design as "modular", but it is unclear what aspect of the reaction network can be considered modular. My understanding of modularity is being able to swap out different sequences or domains of probes and reporters onto the origami surface, and yielding similar outputs. The authors should discuss this claim thoroughly based on their results.

Response: The modularity of the DNA IC-CLA is evident in its ability to execute distinct algorithms or tasks by simply altering the sequences within the respective computing zones. For instance, we configured our DNA IC-CLA to implement the algorithm $f(x) = 2a + 4b - c - 3d$, utilizing it for classifying serum samples from non-small cell lung cancer (NSCLC) patients and healthy individuals. By merely substituting the probes in the weighing zones, without altering any design principles within the zone, we were able to create a new DNA IC-CLA to execute the algorithm $f(x) = 5a + 3b - 2c - 5d$. This new configuration, while preserving the summation, subtraction, and reporting zones, could be employed to classify novel clinical samples. In essence, the DNA IC-CLA enables the classification of diverse inputs through the reconfiguration of probe/reporter sequences and computing zones, without the need to modify the underlying design rules.

Comment 11: Supplementary Table 4 does not provide any information regarding the conditions of the fluorescence kinetics. Under the Methods section, the authors should specify the concentrations, measurement protocols, and other details regarding the use of the microplate reader.

Response: Detailed experimental conditions and protocols have been added to the current Supplementary Table 5 and the Methods section.

Comment 12: The conclusions section currently describes the general advantages of a molecular classifier for diagnostics. The authors should elaborate on the importance of speed and minimizing crosstalk for DNA IC-CLA. What other implications can such improvements in speed and accuracy provide?

Response: We appreciate the reviewer's insightful feedback. The spatial confinement provided by the DNA origami framework in the DNA IC-CLA enables rapid kinetics in DNA computation and reduces interference between different types of probes positioned in separate, distant zones. This design feature allows us to expedite diagnostic processes by enhancing the classification speed of the DNA IC-CLA. Furthermore, by minimizing interference among various probes, we can decrease the likelihood of misdiagnoses. Additionally, the all-in-one localized DNA chip-based classifier has the potential to advance the field of DNA computation towards applications in cell-based biocomputing, where the entire localized system could be co-delivered to cells for targeted classification. These points have been incorporated into the updated "Discussion" section of the revised manuscript.

Reviewer 3:

This article reported a DNA molecular diagnostic system that is spatially positioned and integrated on DNA origami and is based on the credible SVM linear classifier as the calculation principle. It used multiple miRNA inputs as detection targets, and ultimately achieved faster and more effective cancer diagnosis than traditional free diffusion DNA diagnostic systems. I think this article provides a good idea for the design of future molecular diagnostic systems, which is of great significance at a time when disease diagnosis is required to be faster, more accurate, more intelligent, and more integrated. I recommend publishing after solving the following questions:

Response: We are grateful to the reviewer for taking time to evaluate our work and provide valuable comments. We have fully addressed the reviewer's comments as detailed below.

Comment 1: In page 3, line 76-77, the abbreviation rule of the acronym DNA IC-CLA should be explained by adding underline below specific letters.

Response: We have updated "DNA Integrated Circuits-based Classifier (DNA IC-CLA)" in Abstract section in page 2 and Introduction section in page 3.

Comment 2: I can understand the DNA molecular diagnostic system is based on a SVM linear classifier, but the "neuromorphic architecture" claimed by the authors is not comprehensive. It is recommended to reconsider the usage of the word "neuromorphic" or give clear description about the principle.

Response: A neuromorphic architecture in molecular computation aspires to emulate human cognitive tasks with efficiency and parallelism, leveraging molecular components such as nanoscale transistors, memristors, and other nanoscale devices to construct circuits that mimic the connectivity and functionality of neural networks. Distinct from prior DNA computing devices, which primarily rely on Boolean structures, our DNA IC-CLA is envisioned as a neuromorphic

architecture integrated into a physical chip. This system features several artificial probes, analogous to neurons, each with assigned positions and predetermined weights and threshold values, enabling it to execute arithmetic operations—such as multiplication, addition, and subtraction—for classification purposes. This concept parallels the groundbreaking work published in Nature (Nature 610, 2022, 496–501), which showed a proof-of-principle example of neuromorphic architectures for non-linear classification. Therefore, the result of neuromorphic computation is mainly exhibited as a concept and classification results for diagnostics (Yes/No) instead of detailed mapping each component in the system to the corresponding architecture. We have added some explanations of the DNA IC-CLA's neuromorphic architecture in the results section on page 9.

Comment 3: In Figure 2b and 2d, the concentration of input a, b, c, and d should be noted in the figure legend.

Response: We have added the concentration of input a, b, c and d in the legend of Fig. 2.

Comment 4: In Page 10, line 277-278, “b, Fluorescence kinetics of the multiplication process of non-localized system”, fluorescence kinetics data were not found in Figure 3b.

Response: Fig. 3b should be a bar plot of statistical results. We have revised the figure legend.

Comment 5: The author compared the multiplication computation of non-localized system and DNA IC-CLA under different preparation time in Fig 3b and compared the entire computation in Fig S7. However, only slight difference was observed when comparing the entire computation. Therefore, it is not convincing to claim that the origami-integrated strategy exhibits higher robustness because comparison result of the entire computation is more important for the system. Besides, I wonder why the difference is more obvious in multiplication computation?

Response: Here, we have defined robustness as the measure of computational stability over time. In Figure 3b and previous Fig. S7, we aimed to assess the relative robustness of the DNA IC-CLA and the non-localized system by examining their computational accuracy, as indicated by fluorescence changes ($F_t - F_0$), over a period of days. For instance, the $F_t - F_0$ values of multiplication remained consistent after one week for the DNA IC-CLA, whereas they significantly decreased ($p < 0.05$) for the non-localized system after the same duration, indicating that the DNA IC-CLA exhibits functional stability over time and is thus more robust. We have revised Figure 3b and the current Fig. S11 to provide a clearer presentation. Additionally, Figure 3a illustrates the comparison of kinetic rates between the DNA IC-CLA and the non-localized system during the entire classification process, revealing that the DNA IC-CLA is more than three times faster in kinetics, further emphasizing its superior performance.

Comment 6: In Figure 4d, 4f, S13b, S14c, the author plotted the fluorescence of PCR at cycle 50 or band intensity versus initial miRNA concentrations but used logarithms in horizontal axis. In my opinion, here using logarithms means that the amplification of LATE-PCR is not linear. Then how to avoid the nonlinear deviation caused by such an amplification when performing SVM classification? Besides, names and sequences of target miRNAs used in Figure 4, Figure S13 and Figure S14 should be noted in Figure legend or SI. Error bars should be added in Figure 4 and S13.

Response: We thank the reviewer for pointing this out. With the help of this LATE-PCR, we found a linear correlation of end-point ssDNA products with the logarithm of the target miRNA concentration in the range of 1 to 1000 fM. We therefore processed the RNA-seq data from the database with a similar logarithmic transformation to make the concentration data consistent with the mechanism of LATE-PCR amplification. This logarithmic transformation helps to diminish deviations from both PCR amplification and RNA-seq process, although the deviations cannot be completely avoided in actual situation which is hardly to characterize. We have carefully revised the sentence to be “Therefore, in this step, miRNA can be amplified by LATE-PCR and transformed into ssDNA for subsequent DNA computation without disturbing their original quantity ratios” in page 11.

We have added names of target miRNAs in the legends of Figs. 4, the current Fig. S17 and S18. The sequences of target miRNAs have been added in Supplementary Table 3. Error bars have been added in Figs. 4 and S17.

Comment 7: In line 383, why using $[Ft-F0(FAM)] - [Ft-F0(ROX)] > 0.05$ and $[Ft-F0(ROX)] - [Ft-384 F0(FAM)] > 0.05$ instead of $[Ft-F0(FAM)] - [Ft-F0(ROX)] > 0$ and $[Ft-F0(ROX)] - [Ft-384 F0(FAM)] > 0$? The reason choosing 0.05 should be explained. Besides, what is the proportion of samples entering the range with $[Ft-F0(FAM)] - [Ft-F0(ROX)] < 0.05$ or $[Ft-F0(ROX)] - [Ft-384 F0(FAM)] < 0.05$?

Response: We have established a threshold of 0.05 for classification in order to minimize fluorescence measurement errors and enhance the precision of the classification process. The proportion of synthetic samples and clinical serum samples entering the range $([Ft-F0(FAM)] - [Ft-F0(ROX)] < 0.05$ or $[Ft-F0(ROX)] - [Ft-F0(FAM)] < 0.05$) is about 1/30 and 1/50, respectively.

Comment 8: Units and column names should be added to the table in Figure 5a.

Response: To enhance the clarity of data presentation, the original table featured in Figure 5a has been updated and is now presented as Supplementary Table 1, completed with units and column titles.

Comment 9: The conclusion mentions that the DNA IC-CLA can be reused, but there seems to be no experiment to prove this in the text and SI. Besides, the author also claimed that the design of this work has higher biostability (line 404 of the text), but there is also no experiment to prove this in the text and SI. Please correct here or add corresponding experiments.

Response: We appreciate the reviewer's feedback. Our intention was to convey that the DNA IC-CLA, when assembled from the entire nanostructure, exhibits greater functional stability with respect to small single-stranded (ssDNA) and double-stranded (dsDNA) probes. This means that it can produce consistent outputs over a week's time on the same chip (as shown in Figure 3b and Figure S11). We acknowledge that the terms "reused" and "higher biostability" were not accurately applied, and we have rectified these in page 15 of the revised manuscript.

Comment 10: In Figure S2d, the horizontal axis is displayed incompletely.

Response: We have revised the Fig. S2d.

Comment 11: In line 243 and Figure S8, what is the calibration system? Explanations and

discussions should be added.

Response: The calibration system is designed such that the fluorescence change ($F_t - F_0$) for a sample upon the addition of input c is set to 1, serving as a reference to calibrate the fluorescence changes for the remaining three samples in both the non-localized system and the DNA IC-CLA. We have included detailed explanations and discussions of this calibration process in page 9 of the main text and in the legend of current Figure S12c (formerly Figure S8).

Comment 12: In Figure S8, there seems to be no difference in the fluorescence kinetics between non-localized system and DNA IC-CLA seems? In the $cx1$ group, the non-localized system even showed a slightly higher kinetics, which is conflicted with the discussion in line 215-218 of the main text. It is recommended to test three or more parallel samples to characterize the kinetics.

Response: We thank the reviewer for pointing this out. The previous Figure S8 (current Figure S12) presents weighting operations of samples with addition of inputs of a , b , c and d , respectively, while previous lines 215-218 discussed the fluorescence kinetics of multiplication plus summation cascades. As previously mentioned in our response to the Comment 6 from Reviewer 1, the multiplication operation is executed through a three-step DNA strand displacement reaction (SDR) which does not exhibit substantial kinetic differences between localized and solution states, as observed in both experimental and simulation data. However, the more steps involved in SDR, the more pronounced the reaction speed changes we can achieve with the localized system (e.g. subtraction and the entire computation in Fig. 3a). Therefore, the results of current Figure S12 and Fig. 3a are not contradictory. We have revised the legend of current Figure S12 to avoid misleading.

Comment 13: In Figure S9, different combinations of DNA input concentrations should be given in the figure legend.

Response: We have added different combinations of DNA input concentrations in the current Figure S13 (previous Figure S9).

Comment 14: In Figure S10, it is better to label streptavidin in both red and blue reporting zones with different patterns. What is difference between the 16 imaged origami in the bottom? Different combinations of DNA input? It should be noted in the legend.

Response: We are grateful for the reviewer's constructive comments. The red and blue reporting zones are differentiated by the loop structure of the unfolded M13 DNA segment. In the revised Fig. S15 (formerly Fig. S10), we have added labels indicating the presence of streptavidin in these zones. Both the red and blue reporting zones are immobilized with 18 RE probes (for Inputs a and b) and RF probes (for Inputs c and d), creating 18 distinct binding sites for streptavidin molecules in each zone. Theoretically, 18 streptavidin molecules should be loaded into either the red or blue reporting zone upon the addition of 20 nM of Input a or c , respectively. We have provided six representative atomic force microscopy (AFM) images for each labeling type, corresponding to the addition of Input a or c . However, due to the limitations of the AFM characterization method and sample preparation, some images may not clearly resolve the presence of all 18 streptavidin molecules within the red or blue reporting zones. Therefore, we mainly used the end-point fluorescence to verify the correct reporting. We have added the note in the legend of Fig.S15.

Comment 15: In Figure S11. This result is not convincing. It is recommended to reconstruct the origami imaging or just remove this figure.

Response: We have removed this Figure following the reviewer's suggestion.

Comment 16: In Figure S12, the legend is too concise to be understood. What is FRD? What is the meaning of fold change? Explanations should be given.

Response: We appreciate the reviewer's clarification. FDR refers to the false discovery rate, a statistical measure used to control the expected proportion of false positives among the rejected hypotheses in multiple testing problems. Fold change is employed to determine the relative changes in gene expression levels between cancerous and normal samples, serving as a fundamental approach for identifying differentially expressed genes. We have incorporated an explanation of these terms into the legend of the current Fig. S16 (formerly Figure S12).

Comment 17: In line 327 and Figure S15, what is the concentration of the miRNAs amplification products after LATE-PCR amplification?

Response: We have revised the data presented in the current Fig. S18 (formerly Figure S15). The findings indicate a linear relationship between the concentrations of single-stranded DNA (ssDNA) products and the initial logarithmic concentrations of complementary DNA (cDNA) and microRNA (miRNA). After amplification through LATE-PCR, cDNA and miRNA at concentrations ranging from 1 to 1000 fM yielded ssDNA products in the range of 7 to 24 nM and 4 to 20 nM, respectively, which are within the detectable limits for the DNA IC-CLA.

Comment 18: In Figure S16e, it is better to use percentage as the unit for the vertical axis.

Response: We have changed the unit of vertical axis to percentage in current Fig. S20e (previous Figure S16e).

Comment 19: In Table S4, many characters are not displayed correctly. Please check the table carefully and correct the errors.

Response: We have corrected the errors in current Table S5 (previous Table S4).

Comment 20: In Table S4, in the fluorescence kinetic experiment condition, why choosing 37°C instead of room temperature? Are there any considerations?

Response: Our aim is to demonstrate the capability of the DNA IC-CLA to operate at 37°C, a temperature close to physiological conditions, thereby highlighting its potential for in situ computation within living cells.

Reviewer 5:

This work presents a localized molecular classifier where DNA strand displacement computations are localized on a DNA origami. The authors present each of the computing blocks separately. They show how this localized classifier is faster than having just the same computing elements in bulk. Finally, they show how this classifier, combined with a LATE-PCR amplification can be used in a clinical diagnosis setup.

Globally the work is impressive because it combines 3 different expertise of the DNA nanotechnology field: structural DNA nanotechnology, DNA computing and medical diagnostics. These thematics resonate with a multitude of other fields: material science, computer science, physics, biochemistry and biology. Therefore, this reviewer believes it will be of interest for the broad audience of Nature communications, once the authors address several crucial points.

Addressing those points is necessary to properly assess this work and put it in perspective with other existing techniques as well as future developments in the field.

Response: We extend our gratitude to the reviewer for their time and effort in evaluating our work, and we deeply appreciate the support and constructive feedback provided. We have meticulously revised the manuscript to address the reviewer's comments, as detailed below.

Comment 1: First, one of my major concerns is that it is hard to understand the whole scheme and how the modules are arranged together. The resolution of the figures is low, and the labels of the DNA domains are barely legible. The captions of the Supplementary Figure are terse (usually one nondescript sentence that does not do justice to the figure). The schemes are hard to understand because it is not clear what counts as a reagent and what counts as a product (for instance they write chains of reactions like $A+B \rightarrow C+D \rightarrow E+F$. In this case, is D a reagent for the second reaction or a product of the first reaction? It is unclear. Chemists have conventions for this. In the chain above, C and D would be product of the first reaction, and any additional reagent for the second reagent would be indicated by a curved arrow merging with the main arrow of the reaction). Another related concern is that the method section does not clearly reflect what was done. There is a lot of emphasis on the preparation of the components (assembly, purification...), but little on how the computation was actually done experimentally. Also the authors do not explain how sequences were designed, although the homology of domains seen in strand displacement typically places strong constraints on design.

Response: We are grateful to the reviewer for these detailed comments. We have meticulously revised the schemes and legends of figures in both the main text and supplementary information (e.g., Figure 2 and Fig. S3, S4, S7, and S8). For instance, in the enhanced scheme for multiplication, the main cascades are denoted by straight solid arrows, reactants by curved solid arrows, and waste products by curved dashed arrows. Additionally, we have included high-resolution figures to improve clarity. We trust that these revisions will enhance the presentation. Moreover, we have refined the Methods section to detail the experimental conditions and protocols. The primary staples used to fold the DNA IC-CLA were sourced from the literature (Supplementary Reference 3: Chhabra, R. et al., J. Am. Chem. Soc., 2007, 129, 10304-10305). All DNA probes utilized in the DNA computation have a GC content of 30–70%, featuring short toehold domains and long branch migration domains to prevent secondary structure formation, minimize undesired strand interactions, and facilitate the desired strand displacement reactions. The a_2' domain (17 nt) of H_a is shorter than the a_2 domain (27 nt) of A_2 , which increases the effective strand-displacement reaction rate between H_a and the L_1 -A probe in the presence of Input a, rather than between H_a and M probes. Similarly, H_{b4} , Input c, and H_{d3} preferentially react with L_2 -B, L_3 -C, and L_4 -D probes, respectively, rather than with M probes, thereby reducing leakage. The N probe has 8 nt toeholds compared to the 6 nt toeholds in RE or RF probes, which enhances the effective strand-displacement reaction rate between the N probe and E or F probes, compared to that between RE

or RF and E or F probes, respectively. This allows E and F probes to execute the subtraction calculation first, followed by a catalytic amplification. Lastly, the candidate sequences were validated using NUPACK for binding energy and specificity assessment. We have updated the sequence design details in Supplementary Note 3.

Fig.2 Design and validation of DNA IC-CLA.

Comment 2: Another concern is that the authors tend to emphasize positive results (like correct classification), while avoiding discussing negative results (like the severe background seen in some fluorescence experiments). They should discuss and experimentally characterize this, rather than putting it under the rug. For instance, due to the nature of strand displacement, some strands are likely to bear complementary domains and fleetingly bind, which may cause leakage. For instance the helper strand can bind to the M probes to form a complex that leaves a toehold open for toehold exchange (This is easy to check on Nupack). The authors should better discuss and experimentally measure this.

Response: We appreciate the reviewer's observation and concur with the suggestion that negative results are equally valuable for discussion. The spatial confinement provided by the DNA origami framework in the DNA IC-CLA enables rapid kinetics in DNA computation and reduces interference between different types of probes positioned in separate, distant zones, which contributes to enhanced classification accuracy (Figures 3, 5). We acknowledge the leakage observed in Figure 2 of the DNA IC-CLA. Upon investigation, we identified the source of leakage as primarily stemming from interactions between M probes and M reporters, which are freely diffusible at high concentrations and drive the cascaded strand displacement reaction. This issue could potentially be mitigated by refining the sequence design of the probes, such as by employing the 1 nt-gap design (Lv et al., Nature, 2023, 622, 292). The leakage is unlikely to result from hybridization between the Helper strand and M probe, as suggested by NUPACK predictions. We have incorporated these findings into page 6 of the revised manuscript (Fig. S6).

Comment 3: Assess precisely Performance and Speed of classifier (Experiment needed), including more clearly and fairly input datasets used for classification.

The classification of biological samples is complex because of additional purification steps, but the authors need to be precise about the real difficulty of the classification part: How close are the inputs that should give different outputs? What is the relative difference between these 2 close but different classified inputs?

Response: To evaluate the efficacy of our classifier, we have included new separation margin data (Supplementary Note 2, Fig. S14). Our initial analysis focused on the separation margin of the subtraction operation in DNA IC-CLA, examining various combinations of weighted sums (specifically, E and F probes). The concentration of E and F probes tested ranged from 0 to 50 nM. As depicted in Fig. S14a, the upper-left, diagonal, and lower-right areas correspond to the results for $E < F$, $E = F$, and $E > F$, respectively. The data revealed that signal ambiguity arose when the concentrations of the two weighted sums were too similar, leading to the misclassification of 7 samples (indicated by grey dots). We have delineated the region where $E - F = \pm 10$ nM as the separation margin for subtraction, within which 19 samples were challenging to classify accurately. The remaining 30 samples, which were beyond this margin, produced the expected outcomes. In summary, the findings indicate that DNA-IC CLA can effectively differentiate samples when the absolute difference in E-F exceeds 10 nM.

Drawing from these results, for the function $f(x) = 2 \times c(a) + 4 \times c(b) - 1 \times c(c) - 3 \times c(d)$, we prepared 40 samples with $f(x)$ values approximately centered around 10 nM to validate the experimental classification margin (Fig. S14b). The findings indicated that 20 samples with $f(x)$ values within the range of $f(x) = \pm 10$ nM (marked by grey lines) were prone to misclassification, including 16 incorrectly reported samples (grey dots). Conversely, the other 20 samples that were well beyond this margin were accurately classified. When analyzing synthetic and clinical samples, we employed LATE-PCR to amplify the target sequences, ensuring that the $f(x)$ values remained within the classifiable range.

We have evaluated the classification performance of our in silico-trained classifier using a validation set comprising 270 non-small cell lung cancer (NSCLC) and 27 healthy samples from the TCGA database (Fig. S13c). The classifier output values for 17 samples (5.7%) fall within the range of ± 10 nM (indicated by grey lines), which presents a challenge for accurate classification. Among these, 5 samples (1.7%, represented by grey dots) lie on the incorrect side of the separation line (dotted line) and are deemed impossible to classify correctly. The remaining 280 samples (94.3%) are well outside this margin, suggesting that our classifier is expected to achieve high diagnostic accuracy for NSCLC. The small proportion of samples within the separation margin indicates that our classification model is robust and suitable for high-precision NSCLC diagnosis. To evaluate the speed of our classifier, we conducted a simulation to compare the performance of DNA IC-CLA with non-localized cascades (as illustrated in Fig. S10). The findings indicated that an increased local concentration of probes correlates with a faster reaction rate, and this acceleration is further amplified during the catalytic amplification phase of signal reporting.

Comment 4: More quantitative and precisely compare themselves with state of the art for speed and performance

Response: Despite the increasing number of DNA molecular computation systems reported, these systems vary significantly in circuit size and application objectives. Below, we present a table highlighting some of the most notable works in this field. Generally, systems with fewer cascade layers or localized circuits exhibit superior kinetics (Nature Nanotechnology, 2017, 12(9): 920-927), while larger systems are capable of tackling more complex tasks, such as pattern recognition (Nature, 2018, 559(7714): 370-376). Recently, Okumura et al. developed a DNA-enzyme hybrid-based classifier that performs non-linear decision-making in response to miRNA inputs,

significantly enhancing classification speed (Nature, 2022, 610, 496–501). In terms of diagnostic applications, our previous DNA molecular computation design for cancer diagnosis (Nature Nanotechnology, 2020, 15, 709) required over 6 hours to report results, with the time constraint partially due to computation in free solution. Compared to other bulk molecular methods, this work leverages the DNA origami framework to co-localize computing probes, thereby reducing computation time.

Comment 5: Estimate (orders of magnitude) how/why localized DNA computing is an improvement over other methods.

Response: Compared to nonlocalized systems, our DNA CI-CLA has demonstrated three enhancements or improvements, including (i) the markedly accelerated computation speed (over 3 times faster) of the entire computation process (Figure 3a), (ii) a greater functional stability over time, i.e. the $F_t - F_0$ values remained consistent for the DNA CI-CLA but significantly reduced for the non-localized system after a week (Figure 3b and Fig. S11), and (iii) the potential of this all-in-one localized DNA chip-based classifier being delivered to cells or other entities as an integrated system to facilitate its advanced application in cell-based biocomputing.

Comment 6: Add information to figure Legends in the supplementary material

Response: We have added detailed information to the figure legends throughout the supplementary materials.

Comment 7: Correct typos in the schemes

Response: We have meticulously reviewed and rectified any typographical errors present in the schematics across the main text and supplementary materials.

Comment 8: Add high resolution images

Response: We have provided high-resolution pictures in this revision.

Comment 9: *Design/testing/optimization of modules

The authors did not do a good job of showing step by step how the modules were constructed. I would expect them to do experiments showing step by step how the components interact. Due to the nature of strand displacement, all domains must be present in solution from the start. This creates a potential for auxiliary strands to interact in absence of any input. For instance the helper strand Ha will bind to the M-probe (which can be verified in Nupack), creating a potential for leak. This may explain the strong fluorescent background observed in Figure 2 (b and d). I would like to see experiments that better investigate this leak. They authors could for instance test what happens to the leak in absence/presence of M probes and in absence__presence of Helper strands for the a input. This would help to pinpoint the origin of the fluorescence background.

Response: We appreciate the reviewer's suggestion and observation.. We have meticulously revised the schemes and legends of figures in both the main text and supplementary information (e.g. Figure 2 and Fig. S3, S4, S7, and S8) to clearly show the construction of modules in our DNA IC-CLA. We acknowledge the leakage observed in Figure 2 of the DNA IC-CLA. Upon investigation,

we identified the source of leakage as primarily stemming from interactions between M probes and M reporters, which are freely diffusible at high concentrations and drive the cascaded strand displacement reaction. This issue could potentially be mitigated by refining the sequence design of the probes, such as by employing the 1 nt-gap design (Lv et al., Nature, 2023, 622, 292). The leakage is unlikely to result from hybridization between the Helper strand and M probe, as suggested by NUPACK predictions. We have incorporated these findings into page 6 of the revised manuscript (Fig. S6).

Comment 10: Also the authors do not do a good job of showing the linearity of their module. The operation $c(\text{input}) \times \text{Weight}$ which associates a weight to an input concentration is bilinear. It depends linearly both on the input concentration and the weight. The authors did test the linearity for the weights, but they did not test the linearity for the input concentration. I expect them to test that by doing a titration of the input between 0 nM and say 50 nM. I also expect that linearity will break down at low concentration because of the strong fluorescent background. The authors should plot the final raw fluorescence against the concentration of the titrated input, with a fine graining of the concentrations.

Response: We have conducted a titration experiment with input concentrations ranging from 0 nM to 20 nM, as illustrated in Fig. S5. We chose the operation involving input b multiplied by 4 as a case study. Given that the concentration of DNA IC-CLA was fixed at 5 nM, a maximum of 20 nM input b could interact with the DNA IC-CLA, justifying the titration range from 0 to 20 nM for input b. The results indicated a linear increase in output signal with the concentration of input b. In the lower concentration range of 1 to 5 nM, the linearity was less pronounced, which could be due to sample processing errors, instrumental inaccuracies, or limitations in sensitivity to low input concentrations. We have incorporated these results and discussions into pages 5-6 of the main text and Fig. S5.

Comment 11: *Clarity of scheme

The authors should clarify their mechanistic schemes. It is hard to understand what is going on in

Figure 2a, c, e. I presume that the E and F probes are released in solution in Fig2c (but it is not clear in Figure 1c if the E probe is a product or reagent). But then why are the E and F probes in a different form in Fig2e (they are partially duplexed). And in Figure 2e, I do not understand why the E probe was released by the origami. It was already released in the precedent step. Also what does the diode symbol mean here? And I am confused about the subtraction scheme. Does it happen in solution or on the origami?

Response: We have refined the schematics and legends of Figure 2a, c, e, and Fig. S3 and S8 to enhance the clarity of the mechanistic representation. For instance, now the main cascades are denoted by straight solid arrows, reactants by curved solid arrows, and waste products by curved dashed arrows. In essence, both E and F probes are derived from a summation process (Figure 2c, Fig. S3). Following a winner-take-all computation, which involves subtraction (Figure 2e, left), any surplus E or F will engage with the respective reporters to elicit a fluorescence signal. To amplify this reporting signal, we incorporate a fuel-driven circular amplification reaction within the reporting process (Figure 2e, right). In this context, the diode symbolizes an amplifier, adhering to the convention from circuit design where the diode is a standard representation for an operational amplifier (Op-Amp).

Comment 12: *Why use molecular calculation?

For detection in pM/fM range the authors need to do a PCR. Then what are the advantages of doing a PCR and molecular calculations, versus doing a PCR and analyzing this on a computer? Is the gain in speed? or in sensitivity? specificity? Please comment in the text.

Response: Leveraging DNA IC-CLA as an integrated example of molecular calculation, it becomes feasible to streamline the analysis of samples and the reporting of diagnostic outcomes into a single, integrated process. Given the typically low abundance of miRNA or other nucleic acid-based biomarkers in clinical samples, PCR amplification is indispensable. However, our vision extends to integrating automated equipment with molecular computing systems to enable one-step diagnostics, eliminating the need for manual sample processing and data analysis. We anticipate that the distinctive capabilities of DNA computation, which do not require human intervention for data analysis, will justify future endeavors in developing more robust platforms, such as DNA molecular computation-based in situ analysis and diagnosis. We have added these discussions to “Discussion” section of the revised manuscript.

Comment 13: Speed

What is the real time saving vs. other bulk molecular methods? How will the authors method scale-up for more complex calculations? Will it scale-up favorably if layers are added/tiles glued together?

This reviewer is curious to see what the trade-off is between performance and speed for this work, and how it compares to other DNA classifiers in bulk. The author should be careful in their comparison. Some papers use a digital subtraction, and some authors (including those of the paper here) use a pre-amplification step like LATE-PCR. This reviewer expects a table or a graph (whichever the authors believe more appropriate) comparing this work performance and speed to other molecular classifiers.

Response: Despite the increasing number of DNA molecular computation systems reported, these systems vary significantly in circuit size and application objectives. Below, we present a table highlighting some of the most notable works in this field. Generally, systems with fewer cascade layers or localized circuits exhibit superior kinetics (Nature Nanotechnology, 2017, 12(9): 920-927), while larger systems are capable of tackling more complex tasks, such as pattern recognition (Nature, 2018, 559(7714): 370-376). Recently, Okumura et al. developed a DNA-enzyme hybrid-based classifier that performs non-linear decision-making in response to miRNA inputs, significantly enhancing classification speed (Nature, 2022, 610, 496–501).

In terms of diagnostic applications, our previous DNA molecular computation design for cancer diagnosis (Nature Nanotechnology, 2020, 15, 709) required over 6 hours to report results, with the time constraint partially due to computation in free solution. Compared to other bulk molecular methods, this work leverages the DNA origami framework to co-localize computing probes, thereby reducing computation time. It is feasible to distribute computational modules across multiple DNA origami structures and scale up the DNA IC-CLA through hierarchical assembly or by reconfiguring DNA origami to program cascades. The addressability of DNA origami has been utilized to provide spatial organization for molecular information-processing circuits (Nature Nanotechnology, 2017, 12, 920) and molecular motors (Nature, 465, 2010, 206; Nature Nanotechnology, 2012, 7, 169; Science, 357, 2017, eaan6558), theoretically enabling parallel and scalable computation with a limited set of unique molecules. Moreover, advanced cascade circuits or devices can be programmed through DNA origami reconfiguration (Science Robotics, 2023, 8(77): eadf1511). We anticipate that nano- or micro-robots equipped with advanced bio-information processing and manipulation capabilities will greatly benefit from the reconfigurability of DNA origami.

Comment 14: Why use DNA origami?

If the big difference of this paper is bulk vs. localization, then do we really need DNA origami? What does origami really offer compared to strands on a surface or a ball? What's the point of very fine distance control? Why put all weights on the same origami? How can DNA origami formation/reconfiguration be used for computation in the future?

Localizing computation is a different modality from bulk. It deserves to be compared not only with respect to the current implementation but from a more general point of view by extracting the parameters responsible for the different time scales, performances to see the physical limitations and scaling of Bulk vs. localized on DNA origami.

Response: DNA origami stands out as an ideal candidate for the integration of molecular circuits. In the realm of DNA molecular computing, DNA molecules serve as both inputs and outputs, as well as the building blocks of circuits, making DNA origami particularly advantageous for the precise positioning of probes and circuit elements. Beyond the assembly of integrated molecular circuits, DNA molecules are commonly the analytes in bio-analysis and molecular diagnostics, which makes the development of DNA origami-based molecular circuits even more suitable for downstream applications.

Unlike surfaces or spherical structures, DNA origami enables the precise localization of probes at

the nanoscale, both in terms of position and distance, and allows for the control of the stoichiometric ratio of each probe in a single-pot reaction. By meticulously controlling the distance between probes, we can optimize the reaction efficiency in terms of kinetics. Overcrowding probes can lead to steric hindrance, which reduces interaction efficiency and slows down computation. Conversely, too much distance between probes can decrease the total mass of probes, resulting in a lower output level.

As a proof of concept, we have designed our DNA IC-CLA on a single origami structure, as the area of this rectangular DNA origami is sufficient to position all the necessary probes for our current application. However, it is also possible to distribute the computational module across multiple DNA origami structures and scale up the DNA IC-CLA through hierarchical assembly or by reconfiguring the DNA origami to program cascades. The addressability of DNA origami has been harnessed to provide spatial organization for molecular information-processing circuits (Nature Nanotechnology, 2017, 12, 920) and molecular motors (Nature, 465, 2010, 206; Nature Nanotechnology, 2012, 7, 169; Science, 357, 2017, eaan6558), theoretically enabling parallel and scalable computation with a small, constant set of unique molecules. Furthermore, advanced cascade circuits or devices can be programmed through DNA origami reconfiguration (Science Robotics, 2023, 8(77): eadf1511). We foresee that nano- or micro-robots equipped with advanced bio-information processing and manipulation capabilities will greatly benefit from the reconfiguration of DNA origami.

Comment 15: What are the limitations of the classification method? How can you go about overcoming them?

Response: The limitations of this classification method mainly lie in the decision margins, i.e. it could not discriminate between two similar input combinations (concentration differences less than nanomolar) belonging to different classes. We have carefully discussed the classification performance and limitations of our method in Response to Comment 3 from Reviewer 5. In general, we believe that faster kinetics, less leaks, and sensitive reporting of these molecular classifiers can all help improving the accuracy of classification.

Comment 16: The tables of 4D miRNA concentrations can be replaced by a matrix of relative distance or 2 2D plots showing the distribution of miRNA in the samples. This will allow the reader to have a better grasp of the input data.

Response: We have improved Figure 5a and 5b by incorporating both the calculated and experimental diagnostic results. In the revised version, the raw input data table has been relocated to Supplementary Table 1.

Comment 17: This reviewer did such plots for Figure 5. It shows that the data chosen can be classified using a single miRNA marker, and it is unclear what advantage is brought by multi-input classification in this case. The authors chose to use only 30 samples from the 50 appearing in the supplementary materials. How was this choice made? This reviewer believes that showing the limits of the classifier is necessary to really be able to judge its performance.

Response: The levels of miRNA in individuals are characterized by a remarkable diversity and abundance, and variations in either aspect can significantly impact diagnostic outcomes. Figure 5a

provides illustrative examples of miRNA profiles from cancer patients and healthy individuals. Relying on a single miRNA for classification is inadequate for accurate discrimination and does not enable one-step diagnostics without manual data analysis. Regarding the sample sets, the 30 samples depicted in Figures 5a and 5b are synthetic, designed to emulate the miRNA profiles found in the TCGA database. In contrast, the 50 samples featured in the Supplementary materials are actual clinical samples.

Comment 18: The authors should present more clearly the input datasets and how hard to classify they really are.

Response: We have expanded upon Supplementary Note 2 (Fig. S14) with additional data regarding the separation margin of the DNA IC-CLA.

Our analysis began by examining the separation margin of the subtraction operation within the DNA IC-CLA, focusing on various combinations of weighted sums (specifically, E and F probes). The concentration range tested for E and F probes spanned from 0 to 50 nM. In Fig. S14a, the upper-left, diagonal, and lower-right areas correspond to the results for $E < F$, $E = F$, and $E > F$, respectively. We observed that signal ambiguity arose when the concentrations of the two weighted sums were too similar, leading to the misclassification of 7 samples (indicated by grey dots). We have designated the area where $E - F = \pm 10$ nM as the separation margin for subtraction, within which 19 samples were challenging to classify accurately. The remaining 30 samples that fell beyond this margin produced the expected results. In summary, the findings indicate that DNA-IC CLA can reliably differentiate samples when the absolute difference between E and F exceeds 10 nM.

Building upon these results, we prepared 40 samples with $f(x)$ values approximately centered around 10 nM to validate the experimental classification margin for the function $f(x) = 2 \times c(a) + 4 \times c(b) - 1 \times c(c) - 3 \times c(d)$ (Fig. S14b). The results indicated that 20 samples with $f(x)$ values within the range of $f(x) = \pm 10$ nM (marked by grey lines) were difficult to classify correctly, including 16 samples that were misclassified (grey dots). The remaining 20 samples, which were well beyond this margin, were accurately reported. When analyzing synthetic and clinical samples, we employ LATE-PCR to amplify the target sequences, ensuring that the values of $f(x)$ remain within a classifiable range.

We have evaluated the classification performance of our in silico-trained classifier using a validation set comprising 270 non-small cell lung cancer (NSCLC) and 27 healthy samples from the TCGA database (Fig. S13c). The classifier output values for 17 samples (5.7%) fall within the range of ± 10 nM (indicated by grey lines), which presents a challenge for accurate classification. Among these, 5 samples (1.7%, represented by grey dots) lie on the incorrect side of the separation line (dotted line) and are deemed impossible to classify correctly. The remaining 280 samples (94.3%) are well outside this margin, suggesting that our classifier is expected to achieve high diagnostic accuracy for NSCLC. The small proportion of samples within the separation margin indicates that our classification model is robust and suitable for high-precision NSCLC diagnosis. To evaluate the speed of our classifier, we conducted a simulation to compare the performance of DNA IC-CLA with non-localized cascades (as illustrated in Fig. S10). The findings indicated that an increased local concentration of probes correlates with a faster reaction rate, and this acceleration is further amplified during the catalytic amplification phase of signal reporting.

Comment 19: Fig1

b: Clear, but Could add quantitative info/table of Performance & Speed of Classifiers in Bulk vs Localized.

c: DNA IC-CLA is not a clear acronym and should be defined also in caption. Would add “DNA origami” keyword for people from a broader audience.

d: The color of the stripes on the breadboard suddenly change after the summation probes are added. Why? Also the anchored strand have changed (coloured anchor before, blackish anchor after. Why?

Response: We thank the reviewer for this observation. We have made the necessary enhancements to Figure 1b by including the time required for both strategies in the revised manuscript. DNA IC-CLA, which stands for DNA Integrated Circuits-based Classifier, has been clearly defined in the updated text. Regarding Figure 1d, to distinguish between the products of the summation, where $c_b \times 4 + c_a \times 2$ equals Σ_1 (E probes) and $c_c \times 1 + c_d \times 3$ equals Σ_2 (F probes), we have highlighted Σ_1 and Σ_2 with red and blue stripes, respectively, in the updated figure.

Comment 20: Fig2

a,c: At first glance, strand displacement is not obvious (it could be a polymerization from a template). Showing displaced strands or adding some symbol might help the reader.

e: “catalytic entropy-driven amplification” does not appear clearly in the figure. The reader is referred to Supplementary Fig 5, but the caption in this figure is terse (only one sentence). More details on this step seem crucial.

Response: We have improved Fig. 2a, c, e, and the current Supplementary Fig. 8 (formerly Supplementary Fig. 5).

Comment 21: Fig 3:

a: Great and clear

b: What is tested is not clear from the figure only. (need to read the caption).

c: What is tested is not clear from the figure only. (need to read the caption) Would add “Classification results for varying inputs” or something similar. Table representation is not adequate: Hard to grasp quickly the composition of each sample. Please choose a more graphical representation and put the table with raw datas in SUPMAT.

Response: We have updated Figures 3b-c. Specifically, for Figure 3c, we have substituted the raw input data table with the calculated values of $f(x)$. The original input data can be found in Fig. S13.

Comment 22: Fig. 4:

a: not clear from the figure only. Add title like “LATE PCR amplification”

b: Time in X axis is missing. This is important when assessing the speed of the diagnosis.

c,d,e,f: Great

Response: We have revised Fig. 4a-b according to the reviewer's suggestions.

Comment 23: Fig. 5:

a: The raw data table does not allow us to grasp how the input dataset is organized. Please put the raw data in Supplementary material and add a graphical representation as discussed before (this could be several 2D plots, a distance matrix etc).

b: "Results" in the X axis is not clear.

c-e: Great

d: Great but shouldn't this be the first panel before the current "a"?

Response: We have enhanced Figure 5a, b by incorporating both the calculated and experimental diagnostic results, and have relocated the raw data table to Supplementary Table 1. Regarding the current Figure 5e (previously Figure 5d), it outlines the workflow for processing clinical serum samples and is intended to follow the presentation of results from synthetic samples, as depicted in Figure 5d.

Comment 24: Figure S3:

The complementarity of the helper strands seem wrong and all over the place. For input a, a2' strand is listed. For input b, strands b2, b3 and b4' are listed. For input d, strands d2 and d3' are listed (but the low resolution is frankly painful to read, even at maximum magnification). It seems to me that the helper strands H_a should have a domain a₂ (not a₂') to help displace the long top strand. Same comment for input b and c. Please correct as needed and make those symbols readable.

Response: We thank the reviewer for pointing this out. We have revised the current Fig. S4 (formerly Figure S3). The H_a domain (not a₂) is fully complementary to the a₂'* domain of the long top strand (L₁) and partially complementary to the a₂* domain of the M probe. This design facilitates the displacement of the long top strand and prevents unwanted reactions with the M probe. We have applied the same design strategy to the helper strands for inputs b, c, and d.

Comment 25: Figure S5:

The caption should better describe the figure. I do not understand the first reaction: why are there 4 strands anchored on the origami after the reaction while there was only one before. The new strands do not seem to come from the catalysis cycle since none of them bear a red domain. Overall the scheme is hard to understand (not least because the domains are not legible

Response: We have revised the current Fig. S8 (formerly Figure S5). There are 18 RE probes and RF probes on the reporting zones, respectively.

Comment 26: L26: “in a faster and more effective manner” => Be more precise

Response: We have revised the abstract: We demonstrate that the DNA IC-CLA enables accurate cancer diagnosis in a faster (report in about 3 h) and more effective manner in synthetic and clinical samples compared to those of the traditional freely diffusible DNA circuits.

Comment 27: L62: “Slow kinetics” => How slow?

Response: Leveraging the updated simulation and experimental data, we have determined that the DNA IC-CLA operates over 3 times more swiftly for the entire computation process when compared to non-localized systems.

Comment 28: L68: “ideal scaffolds” => Why “ideal”?

Response: The addressable and programmable nature of DNA origami enables the precise localization and arrangement of DNA probes, including but not limited to controlling their position, distance, and stoichiometric ratio. This versatility makes DNA origami an ideal scaffold for a wide array of DNA reaction networks. We have incorporated these insights into the revised manuscript.

Comment 29: L77: “spatially localized DNA classifier (DNA IC-CLA)” => I don’t understand the choice of Acronym DNA IC-CLA.

Response: DNA IC-CLA means DNA Integrated Circuits-based Classifier. We have added the definition in Abstract and Introduction of this revised manuscript.

Comment 30: L163: “It is typically difficult to implement negative weights or weighted sums for DNA-based computation” => Please add relevant references

Response: We appreciate the reviewer for this feedback. Implementing negative weights or weighted sums in DNA-based computation is indeed challenging. For instance, Qian et al. employed a dual-rail technique to achieve negative weights, which, however, resulted in a circuit size increase by a factor of two (Nature, 2011, 475, 368-372). We have included pertinent references (References 15 and 39) in the revised manuscript to further elaborate this topic.

Comment 31: L170: “catalytic entropy driven amplification” => Give a short explanation in main text and add more details in the supplementary materials

Response: To improve the clarity, we have rephrased "catalytic entropy driven amplification" to "entropy-driven catalytic amplification." This process involves an entropy-driven catalytic reaction between the E and RE probes, which is facilitated by Fuel E. This reaction releases E, enabling it to participate in additional catalytic cycles and produce more ssDNA (Re probe). A similar design has been applied to the F probe. We have updated the manuscript and the legend of current Fig. S8 (formerly Figure S5) to reflect these changes.

Comment 32: L240: “A successful linear classifier requires cascaded and accurate mathematical operations for effective data classification” => What is hard in such cascading?

Response: To date, various DNA molecular-based classifiers have been developed and have demonstrated their potential as powerful tools for performing complex computations on different substrates for a range of applications. These applications include solving mathematical problems (Science, 2002, 296(5567): 499-502), recognizing intricate patterns (Nature, 2018, 559(7714): 370-376), and diagnosing diseases (Nature Nanotechnology, 2020, 15(8): 709-715; Science Advances, 2022, 8(47): eade0453). However, the development of DNA molecular-based classifiers with advanced functionalities requires sophisticated cascades and minimal leakage, which remain significant challenges in the field. Otherwise, the classification results may lack credibility. Additionally, diffusible cascades typically take hours to complete computations, making them impractical for complex classification tasks. In response to these challenges, we have been developing DNA IC-CLA (Integrated Circuit-Like Assembly) with the aim of reducing leakage and computational time. Our hope is that our work will inspire other researchers in the field to push the boundaries of DNA molecular computation and enable more efficient and effective solutions for complex classification problems.

Reviewer #1 (Remarks to the Author):

See attached document.

Reviewer #1 Attachment on the following page

This study describes a DNA classifier that can perform molecular computation with spatially localized DNA components that enable faster and more effective reactions than in free solution. Using this approach, the authors demonstrate that their new approach provides results for cancer diagnosis in approximately three hours. This is faster compared to the previous DNA molecular computation design in free solution, which took around six hours to provide results (Nature Nanotechnology, 2020, 15, 709).

The authors have addressed many of my concerns and have provided detailed responses to each of my comments. Most of the errors within the paper have also been fixed. I think this paper will be suitable for publication under *Nature Communications* after the authors address the following comments:

- 1) The model that the authors included in the revised version of this study is thoroughly described. The simulated fluorescence kinetics of multiplication and summation in Supplementary Fig. 10 a and b do not seem to fit very well with those based on experiments. Why is this so?
- 2) Moreover, the simulations predict a clear difference in fluorescence kinetics between non-localized and localized. What is it that the simulation predicts that cannot be accomplished within the experiments?

Reviewer #2 (Remarks to the Author):

Reviewer #3 (Remarks to the Author):

Thank you for your reply to my review comments. I am glad to see that you have made reasonable and appropriate answers and modifications to the manuscript in response to my questions and suggestions. I will recommend receiving the current version of the manuscript.

Reviewer #4 (Remarks to the Author):

Reviewer #5 (Remarks to the Author):

This reviewer is mostly satisfied with the additional work performed by the authors and how the manuscript improved.

However, we believe that replies to comments 4, 5,13,14 need to be improved, and the essence of these replies need to be reflected in the manuscript. Moreover, the replies of the authors to comments 16,17,18 were evasive, which fail to satisfy us. Replying correctly to these points should not take too much time as, it doesn't consist in additional experiments nor complex analysis or simulations. Nonetheless it is crucial to allow the reader to fairly judge this work with respect with other work in the field.

1) This reviewer expects the authors to more quantitative and precisely compare themselves with state of the art for speed and performance (Comments 4,5,13)

- comment 5: Data from Supplementary figure 10, 11, 12 is good and convincing but the quantitative analysis is absent. The extraction of speeds from these figures is left as an exercise for the reader. In order to quantitatively compare non-localized and localized computing the authors need to either extract the speed, or a time over threshold from the curves, add it to the figure and comment it precisely in the text.
- comment 4+13: The authors refer to a table that that this reviewer cannot find anywhere. It seems the authors only reply to this reviewer while this reviewer was expecting the authors to add the comments in the manuscript. Moreover, the reply from the authors is far from being quantitative and precise in the comparison. The short review of the literature of comment 13 is interesting but needs to be more quantitative. (over the whole paragraph, the only quantitative fact mentioned is one calculation taking over 6h)

2) The authors should not only reply to this reviewer but add their explanation as a supplementary note and refer to it in the text. (Comment 14)

3) The authors did not correctly reply to comments 16,17 and 18 concerning the input datasets. Although Figure 5 is now clearer there is no faithful representation of the input data for the samples. The samples are always defined after classification. The fact that the input data is not hard to separate is not a problem as long as the authors explain it carefully. The reply to comment 17 has good elements that should appear in the text.

Reviewer #6 (Remarks to the Author):

Response to Reviewers' Comments

Reviewers #2, 4 and 6:

"I co-reviewed this manuscript with one of the reviewers who provided the listed reports. This is part of the Nature Communications initiative to facilitate training in peer review and to provide appropriate recognition for Early Career Researchers who co-review manuscripts."

Response: We thank the reviewers for providing valuable comments to our work.

Reviewer #1 (Remarks to the Author):

This study describes a DNA classifier that can perform molecular computation with spatially localized DNA components that enable faster and more effective reactions than in free solution. Using this approach, the authors demonstrate that their new approach provides results for cancer diagnosis in approximately three hours. This is faster compared to the previous DNA molecular computation design in free solution, which took around six hours to provide results (Nature Nanotechnology, 2020, 15, 709).

The authors have addressed many of my concerns and have provided detailed responses to each of my comments. Most of the errors within the paper have also been fixed. I think this paper will be suitable for publication under Nature Communications after the authors address the following comments:

1) The model that the authors included in the revised version of this study is thoroughly described. The simulated fluorescence kinetics of multiplication and summation in Supplementary Fig. 10 a and b do not seem to fit very well with those based on experiments. Why is this so?

Response: Thank you for your comments and valuable question. In general, the deviation between simulation and experimental results is most likely due to an incomplete purification of DNA IC-CLA in the experiment and a simplified model in the simulation. In specific, although DNA IC-CLA were purified by size exclusion chromatography twice, it was difficult to completely remove the excess unmodified strands that would cause leak reactions. Another possible deviation source is that the simplified model used in simulation might not be able to perfectly reflect all the actual conditions in the experiment. Below shows a more detailed explanation for why the deviation was present in the multiplication step in simulation (Supplementary Fig. S10a).

In the simulation model for multiplication, we streamlined the complex reaction system by considering all probes on the computing platform as a single reactant entity. Additionally, helpers involved in the multiplication reaction were also grouped under the category of reactants. We postulated the following reaction scheme:

This equation suggests that the Input strands are completely converted to Weight strands with the assistance of Helper strands within a certain time frame. However, this transformation is a multi-

step process that involves sequential participation of various Helper strand types. Each step could potentially exceed the anticipated duration, which implies that the reaction rate constant, k_1 , may not accurately capture the reaction dynamics during the initial 60 minutes. This may explain why there is deviation between simulation and experiment in Supplementary Figure 10a.

2) Moreover, the simulations predict a clear difference in fluorescence kinetics between nonlocalized and localized. What is it that the simulation predicts that cannot be accomplished within the experiments?

Response: Thank you for your question. We agree that the distinction in reaction speeds between nonlocalized and localized processes is more evident in our simulation results than in the experimental data presented in Supplementary Figure 10a. Generally, the more steps involved in strand displacement reactions (SDRs), the more pronounced the reaction speed changes we can achieve with the localized system. As Supplementary Figure 10a depicts a three-step SDRs, the speed differences are easily masked by the experimental uncertainties such as incomplete purification. In contrast, our simulation and experimental data align more closely for the summation (a four-step SDRs) and subtraction steps (involving more than five steps), thereby substantiating our assertions regarding the efficacy of the localized system.

Reviewer #3 (Remarks to the Author):

Thank you for your reply to my review comments. I am glad to see that you have made reasonable and appropriate answers and modifications to the manuscript in response to my questions and suggestions. I will recommend receiving the current version of the manuscript.

Response: Thank you for your previous comments and suggestions, which are constructive in enhancing the quality of our manuscript.

Reviewer #5 (Remarks to the Author):

This reviewer is mostly satisfied with the additional work performed by the authors and how the manuscript improved.

However, we believe that replies to comments 4, 5,13,14 need to be improved, and the essence of these replies need to be reflected in the manuscript. Moreover, the replies of the authors to comments 16,17,18 were evasive, which fail to satisfy us. Replying correctly to these points should not take too much time as, it doesn't consist in additional experiments nor complex analysis or simulations. Nonetheless it is crucial to allow the reader to fairly judge this work with respect with other work in the field.

1) This reviewer expects the authors to more quantitative and precisely compare themselves with state of the art for speed and performance (Comments 4,5,13)

- comment 5: Data from Supplementary figure 10, 11, 12 is good and convincing but the quantitative analysis is absent. The extraction of speeds from these figures is left as an exercise for the reader. In order to quantitatively compare non-localized and localized computing the authors need to either extract the speed, or a time over threshold from the curves, add it to the figure and comment it precisely in the text.

Response: We are grateful for the valuable feedback provided by the reviewer. In response, we have quantitatively detailed and compared the maximum reaction rates (V_{max}) for multiplication,

summation, subtraction, and overall computation between the DNA Integrated Circuit - Controlled Logical Assembly (IC-CLA) and the free diffusion system in the main text, as referenced in Figure 3a and described in page 8, the 2nd paragraph.

Fig. 3. a, Scheme and fluorescence kinetics of multiplication, summation, subtraction, and the entire computation of non-localized system and DNA IC-CLA. V_{max} is the slope of the fluorescence kinetics in the first 10 min.

We have put the quantitative comparisons between the localized and non-localized systems in the main text shown below (highlighted in page 8, the 2nd paragraph):

“For three-step DNA strand displacement reactions (SDRs) of weighting and four-step SDRs of summation, DNA IC-CLA showed slightly faster kinetics with 1-1.5 times higher reaction rate (V_{max}) (Fig. 3a). Taking into account that the reaction network yields products at an initial rate that is approximately linear in the first 10 minutes, which is closely mirroring the actual reaction rate of DNA computing cascades, the rate progressively decreases as the reaction continues and the concentrations of the probes fluctuate. The reaction rate (V_{max}) is defined as the slope of the fluorescence kinetics in the first 10 minutes. When more steps of computation such as subtraction (over 5 steps of SDRs) was cascaded, the localized DNA IC-CLA showed a much higher V_{max} (3 times higher).”

Regarding Supplementary Fig. 11, the quantitative analysis results are listed in the Supplementary Fig. 11b. The essential findings from these figures underscore that the computational robustness of the DNA IC-CLA exceeds that of the nonlocalized system. Supplementary Fig. 12a shows the fluorescence kinetics of multiplication operations in non-localized system and DNA IC-CLA. Their end point differences (Fluorescence difference ($F_t - F_0$)) are used to exhibit the computation accuracy with the quantitative comparison shown in Fig. 12b and 12c.

- comment 4+13: The authors refer to a table that that this reviewer cannot find anywhere. It seems the authors only reply to this reviewer while this reviewer was expecting the authors to add the comments in the manuscript. Moreover, the reply from the authors is far from being quantitative and precise in the comparison. The short review of the literature of comment 13 is interesting but needs to be more quantitative. (over the whole paragraph, the only quantitative fact mentioned is one calculation taking over 6h

Response: Thank you for your insightful comments. As highlighted earlier, the design and applications of DNA molecular computation exhibit considerable variation across different studies, which complicates the process of quantitative comparison. Typically, systems that feature fewer

cascade layers or employ localized circuits exhibit superior kinetics. Conversely, more extensive systems, with their increased complexity, are more adept at managing sophisticated tasks like pattern recognition.

To facilitate the convenience of readers, we have curated a selection of key studies in DNA molecular computation in Supplementary Note 3 and Table 3. In addition, the issues you raised have been thoroughly addressed in the Discussion section of our revised manuscript. There, we delve into the subtleties and challenges inherent in benchmarking diverse DNA molecular computation systems."

References	Scale (numbers of computation steps)	Localized or diffused	Time required for reporting	Application
Neural network computation with DNA strand displacement cascades (Nature, 2011, 475(7356): 368-372)	Large	Diffused	8-60 hours	Artificial neurons for a Hopfield associative memory
A spatially localized architecture for fast and modular DNA computing (Nature Nanotechnology, 2017, 12(9): 920-927)	Small	Localized	2-3 hours	Elementary Logic algorithm
Scaling up molecular pattern recognition with DNA-based winner-take-all neural networks (Nature, 2018, 559(7714): 370-376)	Large	Diffused	8-20 hours	Pattern recognition
A molecular multi-gene classifier for disease diagnostics (Nature Chemistry, 2018, 10(7): 746-754)	Medium	Diffused	2 hours for computing, 6 hours for classification	Classification of synthetic samples
Cancer diagnosis with DNA molecular computation (Nature Nanotechnology, 2020,	Medium	Diffused	6 hours	Diagnosis of synthetic and clinical samples

15(8): 709-715)				
Nonlinear decision-making with enzymatic neural networks (Nature, 2022, 610, 496–501)	Small	Diffused	6-16 hours	Non-linear classification
This work	Medium	Localized	3 hours	Diagnosis for synthetic and clinical samples

Supplementary Table 3. Comparison between DNA IC-CLA and other DNA molecular computation systems.

2) The authors should not only reply to this reviewer but add their explanation as a supplementary note and refer to it in the text. (Comment 14)

Response: Thank you for your comments. We have added the comments to the Discussion section of the manuscript (highlighted in page 15, the 2nd paragraph) as shown below:

“DNA origami stands out as an ideal candidate for the integration of molecular circuits. In the realm of DNA molecular computing, DNA molecules serve as both inputs and outputs, as well as the building blocks of circuits, making DNA origami particularly advantageous for the precise positioning of probes and circuit elements. Beyond the assembly of integrated molecular circuits, DNA molecules are commonly the analytes in bio-analysis and molecular diagnostics, which makes the development of DNA origami-based molecular circuits even more suitable for downstream diagnostic applications.”

3) The authors did not correctly reply to comments 16,17 and 18 concerning the input datasets. Although Figure 5 is now clearer there is no faithful representation of the input data for the samples. The samples are always defined after classification. The fact that the input data is not hard to separate is not a problem as long as the authors explain it carefully. The reply to comment 17 has good elements that should appear in the text.

Response: Thank you for your comments. We have summarized the detailed input data for Figure 5 in Supplementary Table S2 and highlighted the related description in the main text.

As for the separation margin of the classifier (comment 18), we have previously offered a detailed explanation in our latest response. To reiterate, the DNA-IC CLA demonstrates a high level of reliability in differentiating samples when the absolute difference between E and F surpasses 10 nM. To enhance the likelihood of successful classification and to validate the performance of the DNA-CLA more stringently, we have designed and included synthetic samples in our experiments that are beyond the ± 10 nM threshold. This approach ensures a thorough test of the system's capabilities. Upon re-examining the clinical database (TCGA), we found that only a small subset, approximately 4%, of clinical samples are categorized within the challenging-to-classify range of ± 10 nM, as indicated by the grey lines in our figures. We've added the related discussion (comment

17) in Supplementary Note 2.

Reviewer #1 (Remarks to the Author):

Thank you for addressing my questions. I am happy with the responses and think the paper is now appropriate to be published. I recommend that the current version is published under Nature Communications.

Reviewer #2 (Remarks to the Author):

Reviewer #5 (Remarks to the Author):

We would like to congratulate the authors for their work. They authors have replied well to all my questions and added a lot of content in the supplementary materials. The addition of Supplementary table 3 and the extraction of quantitative information allows now to compare faithfully this work with the literature. The authors need to refer more precisely to some of this content in the main text as we will point below.

Once these minor adjustments (that we believe does not require an additional round of review) are done we recommend accepting this manuscript for publication.

In their reply to my comment, the authors state more clearly the limitation of their classifier in their rebuttal "re-examining the clinical database (TCGA), we found that only a small subset, approximately 4%, of clinical samples are categorized within the challenging-to-classify range of ± 10 nM". This sentence is important for a reader to understand the limitation of the classifier, and it should go somewhere in the main text when discussing the results of Figure 5, and a reference to Supplementary Figure 14 should be added.

Reviewer #6 (Remarks to the Author):

Response to Reviewers' Comments

Reviewers #2 and 6:

"I co-reviewed this manuscript with one of the reviewers who provided the listed reports. This is part of the Nature Communications initiative to facilitate training in peer review and to provide appropriate recognition for Early Career Researchers who co-review manuscripts."

Reviewer #1 (Remarks to the Author):

Thank you for addressing my questions. I am happy with the responses and think the paper is now appropriate to be published. I recommend that the current version is published under Nature Communications.

Response: We thank the reviewers for providing valuable comments that significantly improve our work.

Reviewer #5 (Remarks to the Author):

We would like to congratulate the authors for their work. They authors have replied well to all my questions and added a lot of content in the supplementary materials. The addition of Supplementary table 3 and the extraction of quantitative information allows now to compare faithfully this work with the literature. The authors need to refer more precisely to some of this content in the main text as we will point below.

Once these minor adjustments (that we believe does not require an additional round of review) are done we recommend accepting this manuscript for publication.

In their reply to my comment, the authors state more clearly the limitation of their classifier in their rebuttal "re-examining the clinical database (TCGA), we found that only a small subset, approximately 4%, of clinical samples are categorized within the challenging-to-classify range of ± 10 nM". This sentence is important for a reader to understand the limitation of the classifier, and it should go somewhere in the main text when discussing the results of Figure 5, and a reference to Supplementary Figure 14 should be added.

Response: Thank you for providing this valuable suggestion. We have added this statement in the main text when discussing the results of Figure 5 (page 12, 2nd paragraph). A reference to Supplementary Figure 14 has been added.